# MemeSense: An Adaptive In-Context Framework for Social Commonsense Driven Meme Moderation

⚠️**DISCLAIMER: This manuscript features memes that some readers may find vulgar/offensive/hateful.**

**Sayantan Adak[1], Somnath Banerjee[1,3], Rajarshi Mandal[1], Avik Halder[1], Sayan Layek[1], Rima Hazra[2], Animesh Mukherjee[1]**

[1] **Indian Institute of Technology Kharagpur**
[2] **Eindhoven University of Technology, Netherlands**
[3] **Cisco Systems**

## Abstract

Online memes are a powerful yet challenging medium for content moderation, often masking harmful intent behind humor, irony, or cultural symbolism. Conventional moderation systems "*especially those relying on explicit text*" frequently fail to recognize such subtle or implicit harm. We introduce MemeSense, an adaptive framework designed to generate socially grounded interventions for harmful memes by combining visual and textual understanding with curated, semantically aligned examples enriched with commonsense cues. This enables the model to detect nuanced complexed threats like misogyny, stereotyping, or vulgarity "*even in memes lacking overt language*". Across multiple benchmark datasets, MemeSense outperforms state-of-the-art methods, achieving up to **35% higher semantic similarity** and **9% improvement in BERTScore** for non-textual memes, and notable gains for text-rich memes as well. These results highlight MemeSense as a promising step toward safer, more context-aware AI systems for real-world content moderation. [1]

## 1 Introduction

Memes have emerged as a powerful form of online expression, where seemingly lighthearted humor can conceal offensive, derogatory, or culturally charged subtexts. Their multimodal nature combining images, text, and symbolism poses significant hurdles for content moderation systems, especially those built primarily around textual analysis Maity et al. (2024); Jain et al. (2023); Jha et al. (2024b;a). Large vision-language models (VLMs), including GPT-4o OpenAI et al. (2024), Gemini 2.0 Team et al. (2024), and Qwen 2.5 Qwen et al. (2025), often show reduced accuracy on image-centric memes precisely because they depend heavily on overt text clues Sharma et al. (2023); Agarwal et al. (2024). In contrast, humans effortlessly parse memes by applying commonsense reasoning and recalling mental examples of similar situations. This can be attributed to the *social commonsense* Naslund et al. (2020); Arora et al. (2023); Office of the Surgeon General (OSG) (2023)[2] capabilities of humans which include *recognizing social norm violations* (e.g., hate speech, body shaming, misogyny, stereotyping, sexual content, vulgarity), *assessing credibility* (e.g., misinformation), *empathy and ethical judgment* (e.g., child exploitation, public decorum and privacy, cultural sensitivity, religious sensitivity), *contextual interpretation* (e.g., humor appropriateness), and *predicting consequences* (e.g., mental health impact, violence, substance abuse). This human-like capacity to interpret subtle or symbolic cues underscores the need for moderation frameworks that can replicate such higher-level reasoning rather than relying purely on text or raw pixels.

Early multimodal models have attempted to fuse vision and language through joint embeddings or cross-attention mechanisms Shin & Narihira (2021); Radford et al. (2021), yet they tend to place disproportionate

---

[1]The code and Dataset are available at: `https://github.com/sayantan11995/MemeSense`
[2]`https://en.wikipedia.org/wiki/Commonsense_reasoning`

emphasis on textual data. As a result, subtle image-based cues – such as historical references, cultural icons, or visually encoded irony – can slip through the cracks Zhang et al. (2024). Detecting such implicit signals requires not just better model capacity, but the ability to interpret content in light of prior socially grounded examples. Inspired by how humans recall similar experiences to contextualize new ones, we explore a retrieval-augmented approach that grounds meme understanding in examples enriched with commonsense and cultural cues. This design enables the model to move beyond literal interpretation and capture the symbolic and contextual signals embedded in multimodal content, especially when explicit textual markers are absent or misleading.

In this paper, we propose an adaptive in-context learning framework – **MemeSense** that synthesizes commonsense knowledge with semantically similar reference images to enhance the interpretation of meme content. Concretely, **MemeSense** retrieves a curated set of analogous memes, each annotated with cultural, historical, or situational context and incorporates these examples into a unified representation alongside the target meme. By embedding human-like commonsense cues directly into the model's input, we effectively steer its latent space toward the pertinent visual and textual signals present in the attached memes. This synergy allows the model to detect subtle or symbolic markers such as ironic juxtapositions, culturally coded imagery, or sarcastic overlays that often evade traditional pipelines. We validate our framework using mid-sized language models (8B-9B parameters) in post-hoc (zero-shot/in-context) settings, without additional fine-tuning, highlighting the practical utility of intervention generation with minimal resource overhead.

**Our contributions are as follows.**

- We develop a unique multi-staged framework to generate intervention for the harmful memes by leveraging cognitive shift vectors which reduce the requirement of demonstration examples during inference.
- We curate a wide-ranging dataset collection that emphasizes subtly harmful or text-scarce memes, filling a crucial gap in moderation research. This dataset lays the groundwork for a deeper exploration of nuanced meme analysis. We make this dataset publicly available for future research.
- Rigorous experiments demonstrate the efficacy of **MemeSense** even for the memes that do not contain any explicit text embedded in them as is usually the case. We obtain respectively 5% and 9% improvement in BERTScore over the most competitive baseline for the *memes with text* and the *memes without text*. Semantic similarity for memes with as well as without text (almost) doubles for **MemeSense** compared to the best baseline.

## 2 Related work

**Visual in-context learning**: In-context learning (ICL) has revolutionized LLM adaptation by enabling task generalization from a few demonstrations Brown et al. (2020), and recent developments have extended this paradigm to multimodal models for vision-language tasks such as visual question answering (VQA) Alayrac et al. (2022). However, ICL in large multimodal models (LMMs) faces challenges like computational inefficiency due to long input sequences and sensitivity to demonstration selection Peng et al. (2024). To address these issues, in-context vectors (ICVs) have been proposed as compact representations that distill task-relevant information, thereby reducing the dependence on multiple demonstrations at inference time Hendel et al. (2023); Todd et al. (2024). Early non-learnable ICVs showed efficiency gains in NLP but struggled with complex multimodal tasks due to the diversity in vision-language inputs Li et al. (2023); Yang et al. (2024). More recent work introduces *learnable* ICVs that dynamically capture task-specific signals, significantly improving VQA performance while lowering computational overhead Peng et al. (2024). These advancements highlight the importance of optimizing latent task representations and refining ICL strategies for improved multimodal reasoning Yin et al. (2024).

**Intervention generation**: Most intervention strategies for online harm have centered around text-based content, focusing on areas like hate speech Qian et al. (2019); Jha et al. (2024a), misinformation He et al. (2023), and general toxic behavior Banerjee et al. (2024); Hazra et al. (2024); Banerjee et al. (2025). In contrast, multimodal content-particularly memes-remains underexplored despite its unique challenges. Counterspeech has shown potential in mitigating online harm Schieb & Preuss, but it often relies on manually

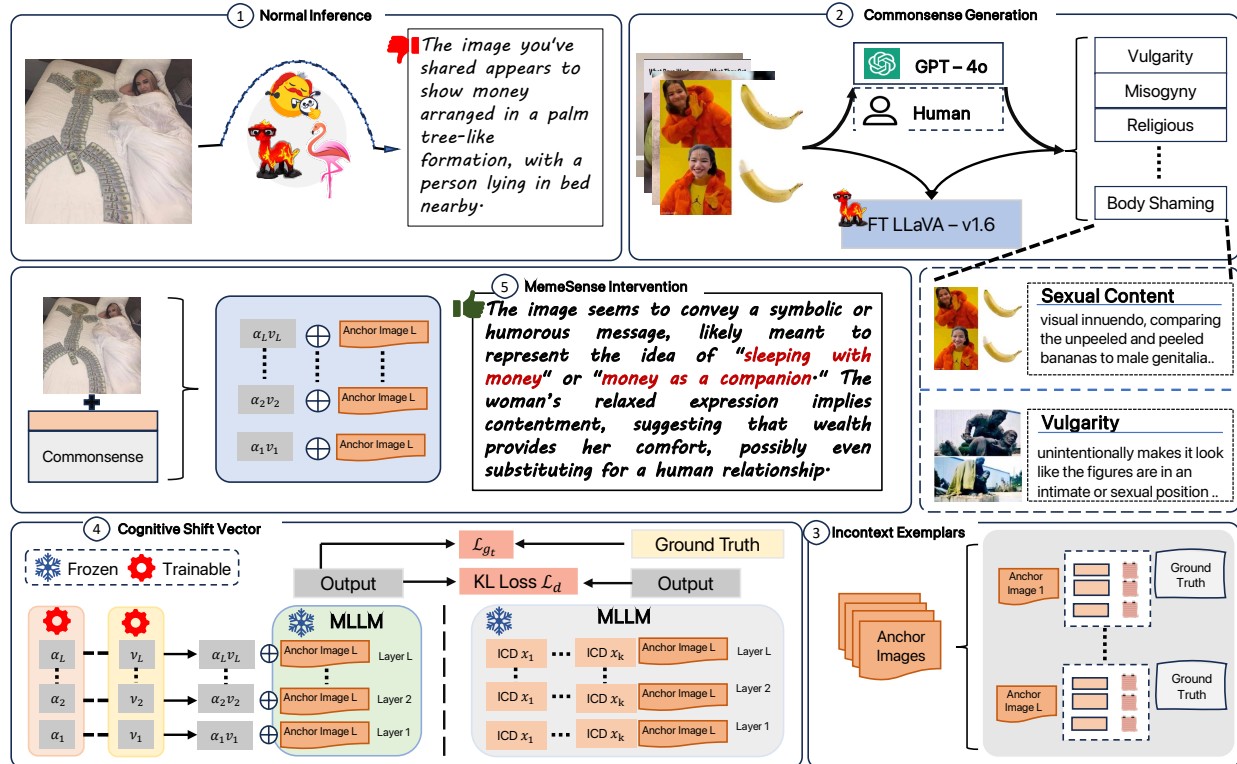

Figure 1: Schematic diagram of MemeSense. Block 1 highlights the challenge of understanding memes in a zero-shot setting using MLLMs. Blocks 2 to 5 illustrate the key stages of our approach: (Block 2) Commonsense Parameter Generation, (Block 3) Exemplar Retrieval, (Block 4) Learning Cognitive Shift Vectors, and (Block 5) MemeSense Inference.

curated responses or supervised datasets Mathew et al. (2018), limiting scalability and adaptability. While advances in LLMs and VLMs Ghosh et al. (2024) have improved automated intervention capabilities, they frequently lack contextual grounding, necessitating knowledge-driven methods Dong et al. (2024). To that end, MemeGuard integrates VLMs with knowledge-ranking mechanisms to enhance meme interpretation and generate more contextually relevant interventions Jha et al. (2024a), marking a step forward in multimodal harm understanding. Similarly, recent efforts Pan et al. (2025); Hee & Lee (2025) employ interpretable, multi-stage reasoning pipelines for harmful meme detection. However, most existing approaches focus primarily on memes with overlaid text, overlooking the challenge of interpreting text-free or visually implicit memes.

## 3 Methodology

In this work, we propose a framework that proceeds in three main stages – (a) **Stage I: Generation of commonsense parameters**: In Stage I, we generate commonsense parameters by instruction-tuning a multimodal large language model (MLLM) to predict contextually relevant insights for each image. (b) **Stage II: Selection of in-context exemplars**: We create a set of anchor images and retrieve corresponding in-context exemplars, which we later use in Stage III. (c) **Stage III: Learning cognitive shift vector**: Finally, we learn a cognitive shift vector by distilling general task information from the exemplars, and then guide the target model to align its representation with the insights derived from these exemplars. The overview of our proposed method is shown in Figure 1.

# 4 Preliminaries

A collection of images is denoted as $\mathcal{IMG}$, where each image $img$ is an item of $\mathcal{IMG}$, i.e., $img \in \mathcal{IMG}$. $GT_{img}$ describes the ground truth intervention on the image. In particular, $GT_{img}$ contains the description about **why the image can/can't be posted on social media?** We consider a set of commonsense parameters $\mathscr{C}$ where $i^{th}$ commonsense parameter is denoted as $c_i \in \mathscr{C}$. A pair consisting of an image and its corresponding commonsense parameters is denoted by $\langle img, \mathscr{C}_{img} \rangle$ where $\mathscr{C}_{img} \subseteq \mathscr{C}$. An image may be associated with multiple commonsense parameters. We partition $\mathcal{IMG}$ into two subsets: **(a)** the training set $\mathcal{IMG}_{tr}$, used at different stages of the training process, and **(b)** the test set $\mathcal{IMG}_{ts}$, reserved for evaluation. The set of training images $\mathcal{IMG}_{tr}$ and test images $\mathcal{IMG}_{ts}$ are disjoint, i.e., $\mathcal{IMG}_{tr} \cap \mathcal{IMG}_{ts} = \emptyset$.

For **Stage I**, we build a training dataset $\mathcal{D}_{\mathscr{C}}$ consisting of images $\mathcal{IMG}_{tr}$ and their respective ground truth image description with commonsense parameters. We represent a fine-tuned vision language model with dataset $\mathcal{D}_{\mathscr{C}}$ as $\mathcal{M}_{\mathscr{C}}$. Further in **Stage II**, we construct an in-context (IC) learning set $\mathcal{D}_{\mathcal{IC}}$ (involves only images from $\mathcal{IMG}_{tr}$ set) to utilize in **Stage III** (see Section 4.3). Each instance in $\mathcal{D}_{\mathcal{IC}}$ is a tuple consisting of $\langle img_a, IC_{img}, GT_{img_a} \rangle$ where $IC_{img}$ is the set of retrieved in-context examples of an anchor image $img_a$. Each in-context example consists of an image $img \neq img_a$, $\mathscr{C}_{img}$, $GT_{img}$. We define the cognitive shift vector set as $\mathcal{CSV}$ and the coefficient set as $\alpha$. In **Stage III**, we use an instruction following MLLM as the target model ($\mathcal{M}$) to further generate the intervention defined as $\mathcal{M}_{ivt}$.

## 4.1 Stage I: Commonsense parameters

In this stage, we aim to fine-tune a vision-language model to produce relevant commonsense parameters for meme images. These parameters represent broad conceptual categories that help assess whether an image is *harmful*, *offensive*, or *inappropriate*, as discussed in Arora et al. (2023); Office of the Surgeon General (OSG) (2023); Gongane et al. (2022). To create the training set $\mathcal{D}_{\mathscr{C}}$, we first use GPT-4o to automatically obtain commonsense parameters for $img \in \mathcal{IMG}_{tr}$ and then perform manual corrections. We employ two expert reviewers experienced in meme moderation to verify the generated commonsense parameters and the intervention from the `GPT-4o`. We provide them with a short description of

Figure 2: Representative example of a harmful meme and the annotated commonsense parameters along with intervention.

each commonsense category (Similar to the Commonsense parameters mentioned in the Table 18) along with one selected meme within each category for reference. They are allowed to update, delete, or add categories based on their judgment. Although we engage two experts due to the niche nature of the task and resource constraints, we ensure high quality through consensus-based evaluation. Notably, their assessment reveals a strong consensus (96% agreement) with the GPT-4o outputs. Lastly, for the fine-tuning phase, we choose an open-weight model instead of a proprietary system such as GPT-4o, given the higher resource demands and stricter usage limitations. We employ the prompt mentioned in Table 18 in Appendix to extract commonsense parameters from GPT-4o.

The example of commonsense parameters $\mathscr{C}_{img}$ of an image $img$ is illustrated in Figure 2. By instruction tuning with this dataset $\mathcal{D}_{\mathscr{C}}$, we obtain the final vision-language model $\mathcal{M}_{\mathscr{C}}$, which we then use at inference time to produce commonsense parameters for new meme images.

---

**Prompt for instruction tuning commonsense generation module**

[INST] <image>*Describe the meme and assess whether the meme has the potential to be perceived as vulgar, harmful, or unethical using specific commonsense parameters.* [\INST]

---

## 4.2 Stage II: Selection of in-context exemplars

In this stage, our objective is to create an in-context dataset $\mathcal{D}_{IC}$ that provides exemplars to guide the latent space of the target model in **Stage III**. To accomplish this, we reuse the training images $\mathcal{IMG}_{tr}$ and, following the authors in Chen et al. (2025); Peng et al. (2024); Qin et al. (2024), treat each image $img \in \mathcal{IMG}_{tr}$ as an anchor. We denote an anchor image as $img_a$. We then select $k$ in-context examples from $\mathcal{IMG}_{tr} \setminus img_a$ using multiple strategies. First, we randomly sample $k$ candidate images to construct the set $IC_{img}$ for each anchor. Apart from random selection, we also leverage semantic retrieval techniques that consider commonsense parameters, image representations, or a combination of both. The detailed setup of in-context retrieval is given in Section 6.

## 4.3 Stage III: Learning cognitive shift vectors

In this stage, the aim is to learn the trainable shift vector set $\mathcal{CSV}$ and coefficient set $\alpha$ so that the target model can generate proper intervention given a meme $img$. We initialize a set of shift vectors $\mathcal{CSV} = \{csv^1, csv^2, \ldots, csv^L\}$ where each shift vector $csv^\ell$ corresponds to each layer $\ell \in L$ in the target model $\mathcal{M}$. $L$ represents the number of layers in target model $\mathcal{M}$. Further, we consider a set of coefficients $\alpha = \{\alpha^1, \alpha^2, \ldots, \alpha^L\}$ which regulate the impact of these cognitive shift vectors across different layers in $\mathcal{M}$. After applying cognitive shift vector set $\mathcal{CSV}$ and $\alpha$ to the model $\mathcal{M}$, we obtain the final model as expressed in Equation 1.

$$\mathcal{M}_{ivt}^\ell = \mathcal{M}^\ell + \alpha^\ell \cdot csv^\ell, \tag{1}$$

Following task analogies from Huang et al. (2024); Peng et al. (2024), our objective is to align the output of $\mathcal{M}_{ivt}$ with the output obtained by including $IC_{img}$ in model $\mathcal{M}$ for a given anchor image $img_a$. To achieve this, we minimize the KL divergence between the output distribution of $\mathcal{M}_{ivt}(img_a)$ and output distribution of $\mathcal{M}$ with IC exemplars $IC_{img}$ for the anchor image $img_a$. The computation of $\mathscr{L}_{od}$ is given in Equation 2.

$$\mathscr{L}_{od} = KL\left(P(img_a|IC_{img}; \mathcal{M}) \parallel P(img_a|\mathcal{M}_{ivt})\right) \tag{2}$$

where $P(img_a|IC_{img}; \mathcal{M})$ and $P(img_a|\mathcal{M}_{ivt})$ represent the output distribution of models $\mathcal{M}$ and $\mathcal{M}_{ivt}$ respectively for anchor image $img_a$.

Further we compute the intervention loss ($\mathscr{L}_{ivt}$) to make sure that the output of final model $\mathcal{M}_{ivt}(img_a)$ is aligned with the ground truth $GT_{img_a}$ (see Equation 3)

$$\mathscr{L}_{ivt} = -\sum_{|\mathcal{D}_{IC}|} \log P(img_a|\mathcal{M}_{ivt}) \tag{3}$$

We compute the final loss as given in Equation 4. $\gamma$ serves as a hyperparameter that determines the relative importance of output distribution loss and intervention loss.

$$\mathscr{L} = \mathscr{L}_{od} + \gamma \cdot \mathscr{L}_{ivt} \tag{4}$$

## 5 Datasets

To advance research on harmful meme intervention, we construct a novel dataset of implicitly harmful memes, sourced from various online social media platforms, including Facebook, Twitter, Instagram, and WhatsApp. Unlike existing datasets that primarily focus on memes with explicit textual content embedded in them, our dataset specifically targets memes that are implicitly harmful or lack embedded text (see Figure 3 for details). These cases pose additional challenges for AI models, as they require nuanced reasoning beyond surface-level textual analysis. Below, we detail our data collection and annotation process.

Table 1: Distribution of various commonsense attributes.

| Commonsense category (meta) | Commonsense parameters | # Memes |
|---|---|---|
| Recognizing social norm violations | *Hate speech* | 23 |
| | *Body shaming* | 74 |
| | *Misogyny* | 51 |
| | *Stereotyping* | 32 |
| | *Sexual content* | 105 |
| | *Vulgarity* | 135 |
| **Assessing credibility** | *Misinformation* | 4 |
| Empathy and ethical judgements | *Child exploitation* | 12 |
| | *Public decorum & Privacy* | 72 |
| | *Cultural sensitivity* | 60 |
| | *Religious sensitivity* | 14 |
| **Contextual interpretation** | *Humor appropriateness* | 251 |
| Predicting consequences | *Mental health impact* | 38 |
| | *Violence* | 43 |
| | *Substance abuse* | 7 |

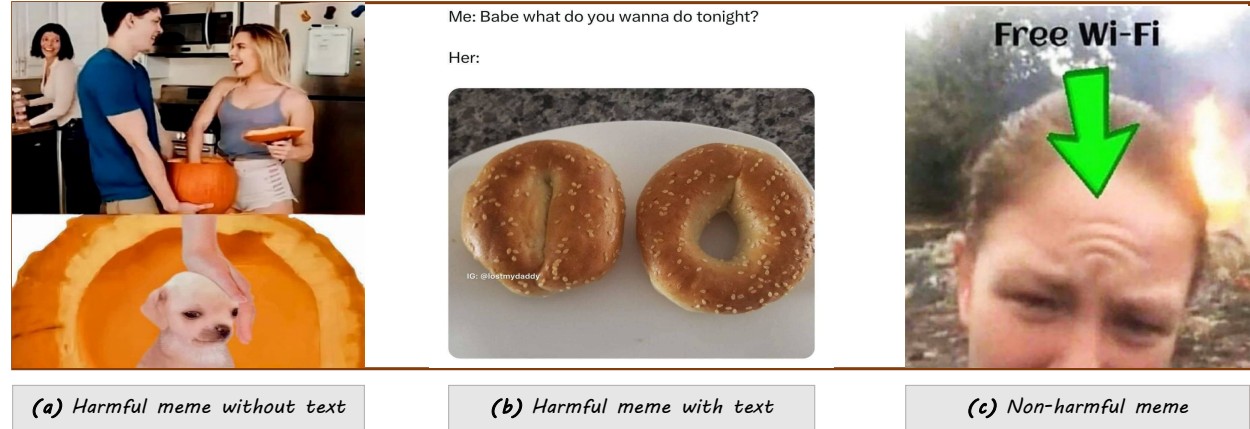

| (a) Harmful meme without text | (b) Harmful meme with text | (c) Non-harmful meme |

Figure 3: Memes can manifest harm in different ways, some rely solely on imagery to convey implicit messages, while others reinforce harm through accompanying text. This figure illustrates the three primary categories: **(a) harmful memes without text**, **(b) harmful memes with text**, and **(c) non-harmful memes**. Prior moderation efforts have disproportionately focused on text-based harmful memes, often overlooking the nuanced and context-dependent nature of purely visual memes.

**Data collection**: We curate memes from publicly available online sources, including Facebook meme pages[3], Twitter adult meme pages[4], public WhatsApp groups, and Instagram meme accounts[5]. In addition, we incorporate phallic[6]-themed memes[7] which may not appear overtly harmful at first glance but can carry implicit harmful implications when shared publicly. Our data collection process resulted in a total of 785 memes.

**Filtering and annotation**: To determine whether each meme exhibits potential harm, we instruct two undergraduate annotators to independently label each meme as either **harmful** or **non-harmful**. We define a meme as harmful if it aligned with any of the 15 predefined commonsense harm categories (e.g., vulgarity, body-shaming), as listed in Table 1. To ensure consistency, we provide the annotators with a concise annotation guideline that includes definitions of each category and representative examples of both harmful and non-harmful memes. We adopt a conservative filtering approach, retaining only those memes that both annotators independently label as harmful. This process results in a final curated dataset of **484 harmful memes**. We calculate Cohen's kappa score, which yields a value of 0.82, indicating strong inter-annotator agreement.

**Training and Test Split:** We allocate **300 memes** for training: Stage 1 involves learning to align annotated descriptions with their corresponding commonsense parameters, while the same set is used to train the Cognitive Shift Vectors (CSV), using the meme content and harm categories as input, and annotated interventions as output. The remaining **184 memes** form the test set, comprising **133** with overlaid text and **51** without.

Once we finalize the harmful meme set, we use GPT-4o along with manual post-processing to generate the corresponding commonsense parameters and ground truth intervention statements, as described in Section 4.1. Figure 3 showcases representative examples from the curated dataset. While our final curated dataset comprises 484 carefully annotated harmful memes, it spans a rich and diverse set of 15 commonsense categories. This breadth ensures strong coverage across varied meme types and contexts. Moreover, our multi-stage framework is specifically designed for adaptability in low-resource settings, allowing flexibility to incorporate additional harmful memes with minimal retraining.

---

[3]https://www.facebook.com/doublemean

[4]https://x.com/DefensePorn

[5]https://www.instagram.com/stoned_age_humour

[6]https://en.wikipedia.org/wiki/Phallus

[7]https://humornama.com/memes/penis-memes/

**Additional ICMM data** In addition to our curated dataset, we also consider the publicly available *Intervening Cyberbullying in Multimodal Memes* (ICMM) dataset Jha et al. (2024a) for evaluation of our approach. This dataset consists of 1000 cyberbullying memes along with their corresponding crowdsourced interventions. After filtering out the corrupted images, we obtain a set of 985 memes along with their ground truth interventions.

## 6 Experimental setup

This section discusses the different experimental configurations of **MemeSense**.

### 6.1 Baselines

For baselines involving zero-shot prompting and in-context learning (ICL), we leverage the same aligned MLLMs used in **MemeSense** – for intervention generation.

**(1) MemeGuard** Jha et al. (2024a): We adapt MemeGuard, a state-of-the-art model for harmful meme intervention generation, as one of our primary baselines. The method operates in two stages involving both vision-language and language models. In our implementation, we use the same base vision-language model (`llava-1.6-mistral-7b-hf`) as in **MemeSense** for the first stage, and `idefics-2-8b-base` for the second. Given a meme, the first-stage VLM generates five descriptive responses intended to capture potential harmful cues. While the original MemeGuard framework incorporates OCR-extracted text as an additional input modality, we omit OCR in our setup to ensure applicability to both text-rich and text-free memes. To remove irrelevant or noisy responses, we compute the semantic similarity between the meme and each generated sentence, retaining only those with a similarity score above 0.2 (a threshold chosen based on manual inspection). In the final stage, the second VLM uses the input meme and the filtered descriptions to generate the corresponding intervention.

**(2) MemeMQA (Modified)** Agarwal et al. (2024): We extend the MemeMQA framework for intervention generation by removing its target identification module and repurposing its explanation generation module. Originally designed to identify targets in hateful memes and explain predictions, MemeMQA now directly generates interventions. This baseline adopts a dual-model architecture, comprising – **(1)** a VLM for rationale generation, same as the base VLM for **MemeSense** and **(2)** a `T5-large` model for intervention generation. The rationale generation VLM is fine-tuned for one epoch with a batch size of 4 and a learning rate of $5 \times 10^{-5}$.

**(3) Commonsense-enhanced prompting**: Given a meme and its automatically generated commonsense parameters, the VLM (same base VLMs as those for **MemeSense**) is instructed to generate an intervention.

**(4) In-context learning (ICL)** Zeng et al. (2024): For a given target meme, we select $k$ ($\in \{1, 2, 4, 8, 10\}$) demonstration examples from the training set, including their annotated commonsense, and provide them as context before prompting the VLM to generate an intervention. For the selection of in-context examples, we use random and semantic retrieval techniques similar to **Stage II** (Section 4.2).

### 6.2 **MemeSense** framework

Recall that **MemeSense** consists of three major stages leveraging (I) multimodal LLMs for generation of commonsense parameter, (II) in-context exemplars selection and (III) subsequent learning of the cognitive shift vector for the **intervention generation**.

For the **Stage I**, we utilize the `llava-v1.6-mistral-7b-hf`[8] model, fine-tuned with QLoRA Dettmers et al. (2023) over 10 epochs using a batch size of 16 and a learning rate of $2 \times 10^{-4}$, with weight decay for optimization.

For the **Stage II**, We employ various strategies for selecting in-context exemplars, detailed as follows:

*Commonsense-based retrieval*: For each predefined commonsense parameter, we select up to five instances from our training set to form a lookup set. Given an anchor image *img* and its corresponding annotated commonsense parameters, we iteratively retrieve at least one instance per parameter to construct the $k$ demonstration examples.

---

[8]`https://huggingface.co/llava-hf/llava-v1.6-mistral-7b-hf`

***Image-based retrieval***: For a given anchor image $img$, we retrieve $k$ demonstrations by computing their semantic similarity with $img$ from the training subset. To achieve this, we first encode all images into dense vector representations using the `CLIP-ViT`[9] multimodal embedding model. When an anchor image is provided as a query, we map it into the same vector space, enabling an efficient similarity search. We then perform Approximate Nearest Neighbor (ANN) Wang et al. (2021) search to identify the top $k$ most similar images. Their corresponding commonsense parameters and ground truth interventions are retrieved as in-context examples, ensuring a contextually relevant selection.

***Combined retrieval***: We also experiment with constructing the $k$ in-context demonstrations by combining the above two approaches. Here, we select $c$ instances from the commonsense based retrieval and $(k - c)$ instances from the image-based retrieval, where $c \in \{1, 2, 4\}$.

For **Stage III**, we primarily employ the `idefics2-8B-base`[10] model to learn cognitive shift vectors and perform inference. In addition, we explore `idefics-9B`[11] and `OpenFlamingo`[12] for intervention generation. The number of in-context demonstration examples is one of $\{1, 2, 4, 8, 10\}$, maintaining a fixed batch size of 2. The shift vector undergoes training for 10 epochs to ensure effective adaptation and we choose $\gamma$ as 0.5.

## 6.3 Evaluation metrics

To rigorously assess the quality of generated interventions, we employ a diverse set of evaluation metrics spanning semantic similarity, lexical accuracy, and readability. Semantic metrics such as BERTScore Zhang* et al. (2020) and semantic cosine similarity Rahutomo et al. (2012) measure the alignment between generated and reference interventions in embedding space. Lexical metrics, including ROUGE-L Lin (2004) and BLEU-4 Papineni et al. (2002), evaluate surface-level text overlap and n-gram precision. Further, we measure readability using the **Flesch Reading Ease Score**, implemented via the `textstat` Python library[13], ensuring the interventions are not only accurate but also coherent and accessible. This holistic evaluation framework enables a nuanced assessment of intervention effectiveness across multiple linguistic dimensions. We use `RoBERTa-large` model for computing BERTScore, and `all-MiniLM-L6-v2` from the *SentenceTransformers* library to compute semantic similarity.

# 7 Results

We structure our experimental results into three key sections. First, we present insights derived from our dataset, highlighting key patterns and observations. Next, we evaluate the performance of our framework on the ICMM dataset, examining its effectiveness in generating interventions. Finally, we delve into a detailed breakdown of performance across different commonsense meta-categories, offering a deeper understanding of the model's strengths and limitations in various contexts.

**Result for our dataset** In Tables 2 and 3, we compare the performance of our framework, MemeSense, with various baselines on memes without text and memes with text, respectively. Across both settings, MemeSense (combined) consistently achieves the highest values for BERTScore (0.91), semantic similarity (0.71 for the memes without text, 0.78 for text-based memes), and ROUGE-L (0.35 and 0.37, respectively), demonstrating its superior capability in generating semantically meaningful and contextually appropriate responses. Among the baseline methods, commonsense-anchored ICL performs competitively but lags behind MemeSense, particularly in terms of semantic similarity score, highlighting the importance of hybrid reasoning strategies.

For memes without text, direct prompting methods struggle with low semantic similarity ($\leq 0.3$), while MemeSense (combined) significantly outperforms them (semantic similarity = 0.71).

---

[9]`sentence-transformers/clip-ViT-B-32`
[10]`https://huggingface.co/HuggingFaceM4/idefics2-8b-base`
[11]`https://huggingface.co/HuggingFaceM4/idefics-9b`
[12]`https://huggingface.co/openflamingo/OpenFlamingo-9B-vitl-mpt7b`
[13]`https://pypi.org/project/textstat/`

Table 2: Result for memes without text. **SeSS**: semantic similarity. * indicates statistically significant improvement from `MemeGuard` and `MemeMQA` using *Mann-Whitney U test* with $p < 0.05$.

| Method | BERTScore (F1) | SeSS | Readability | ROUGE-L (Avg) | BLEU-4 (Avg) |
|---|---|---|---|---|---|
| Direct prompting | 0.81 | 0.27 | **53.36** | 0.05 | 0.001 |
| Direct prompting (w. commonsense) | 0.81 | 0.30 | 21.55 | 0.05 | 0.002 |
| Random ICL | 0.87 | 0.49 | 35.06 | 0.19 | 0.01 |
| Image anchored ICL | 0.86 | 0.41 | 36.49 | 0.17 | 0.02 |
| Commonsense anchored ICL | 0.88 | 0.46 | 34.12 | 0.18 | 0.02 |
| MemeMQA | 0.86 | 0.51 | 52.86 | 0.08 | 0.008 |
| MemeGuard | 0.82 | 0.35 | 51.69 | 0.09 | 0.005 |
| MemeSense (random ICL) | 0.90* | 0.68* | 46.22 | 0.34* | 0.07* |
| MemeSense (image anchored ICL) | 0.90* | 0.70* | 45.57 | 0.35* | 0.08* |
| MemeSense (commonsense anchored ICL) | 0.91* | 0.70* | 45.65 | 0.35* | **0.09*** |
| MemeSense (combined) | **0.91*** | **0.71*** | 44.07 | **0.35*** | 0.08* |

Table 3: Result for memes with text. **SeSS**: semantic similarity. * indicates statistically significant improvement from `MemeGuard` and `MemeMQA` using *Mann-Whitney U test* with $p < 0.05$.

| Method | BERTScore (F1) | SeSS | Readability | ROUGE-L (Avg) | BLEU-4 (Avg) |
|---|---|---|---|---|---|
| Direct prompting | 0.81 | 0.35 | **54.59** | 0.04 | 0.001 |
| Direct prompting (w. commonsense) | 0.80 | 0.28 | 22.02 | 0.04 | 0.001 |
| Random ICL | 0.86 | 0.52 | 31.94 | 0.18 | 0.02 |
| Image anchored ICL | 0.87 | 0.49 | 31.52 | 0.18 | 0.02 |
| Commonsense anchored ICL | 0.88 | 0.55 | 33.25 | 0.19 | 0.03 |
| MemeMQA | 0.86 | 0.54 | 50.28 | 0.10 | 0.009 |
| MemeGuard | 0.84 | 0.39 | 36.36 | 0.09 | 0.004 |
| MemeSense (random ICL) | 0.91* | 0.77* | 46.64 | 0.36* | 0.08* |
| MemeSense (image anchored ICL) | 0.91* | 0.77* | 44.33 | 0.35* | 0.07* |
| MemeSense (commonsense anchored ICL) | 0.91* | 0.78* | 48.74 | **0.38*** | **0.09*** |
| MemeSense (combined) | **0.91** | **0.78*** | 43.38 | 0.37* | 0.08* |

> We want to emphasize that MemeSense achieves **35% improvement in SeSS score and 9% in BERTScore over *MemeGuard***, and **20% improvement in SeSS score and 5% in BERTScore over *MemeMQA*** which are the state-of-the-art methods.

These improvements highlight the effectiveness of our adaptive approach in reasoning about complex memes without having textual cues. Similarly, for memes with text, MemeSense achieves notable improvements in both semantic alignment and lexical overlap (BLEU: 0.08 - 0.09), reflecting its ability to effectively integrate commonsense and image-grounded reasoning. Overall, these results demonstrate that the MemeSense (combined) approach integrating image-anchored, and commonsense-anchored in-context learning (ICL), effectively enhances reasoning and interpretation across different meme types.

Table 4: Comparing the result for the ICMM dataset using the actual model used by MemeGuard in different stages. * indicates statistically significant improvement from `MemeGuard` and `MemeMQA` using *Mann-Whitney U test* with $p < 0.05$.

| Method | BERTScore (F1) | SeSS | Readability | ROUGE-L (Avg) | BLEU-4 (Avg) |
|---|---|---|---|---|---|
| Direct prompting | 0.8 | 0.15 | 67.02 | 0.03 | 0.001 |
| Direct prompting with commonsense | 0.8 | 0.14 | 52.34 | 0.03 | 0.004 |
| Random ICL | 0.82 | 0.16 | 19.63 | 0.09 | 0.005 |
| Image anchored ICL | 0.82 | 0.2 | 22.16 | 0.1 | 0.006 |
| Commonsense anchored ICL | 0.84 | 0.22 | 25.38 | 0.1 | 0.006 |
| MemeMQA | 0.85 | 0.24 | 54.45 | 0.1 | 0.007 |
| MemeGuard (our setting) | 0.79 | 0.18 | 34.45 | 0.04 | 0.001 |
| MemeSense (random ICL) | 0.84 | 0.18 | 44.03 | 0.11 | 0.007 |
| MemeSense (image anchored ICL) | 0.85 | 0.25 | 42.79 | 0.1 | 0.007 |
| MemeSense (commonsense anchored ICL) | 0.86* | 0.27* | 42.22 | 0.11* | **0.009 *** |
| MemeSense (combined) | **0.87*** | **0.31*** | 45.57 | **0.11*** | 0.008 * |

**Result for ICMM data** In Table 4, we show the result of various baselines and compare them with MemeSense for the ICMM dataset. Direct prompting achieves the highest readability (67.02) but performs poorly in semantic alignment (SeSS = 0.15, ROUGE-L = 0.03, BLEU = 0.001), while adding commonsense

knowledge reduces readability further (52.34) without improving semantic scores. In-context learning (ICL) methods, including random, image-anchored, and commonsense-anchored ICL, improve semantic similarity (0.16-0.22) and ROUGE-L (0.09-0.1) but suffer from significantly lower readability (19.63-25.38). Among meme-specific baseline models, **MemeMQA** performs best (SeSS = 0.24, readability = 54.45) as it requires explicit training, while **MemeGuard** underperforms across all metrics (SeSS = 0.18, readability = 34.45). **MemeSense** outperforms all baselines, with **MemeSense** (commonsense anchored ICL) achieving strong semantic alignment (SeSS = 0.27), while **MemeSense** (combined) emerges as the best overall method with the highest BERTScore (0.87) and SeSS (0.31), reasonable readability (45.57), and competitive ROUGE-L (0.11) and BLEU (0.008) scores. This suggests that structured multimodal approaches, particularly **MemeSense** (combined), provide the best balance between semantic coherence and fluency, making it the most effective meme intervention generation strategy.

Table 5: Meta category-wise evaluation results.

| Meta category (Commonsense) | BERTScore (F1) | SeSS | ROUGE-L (Avg) |
|---|---|---|---|
| Contextual interpretation | 0.91 | 0.78 | 0.37 |
| Empathy and ethical judgements | 0.90 | 0.75 | 0.33 |
| Predicting consequences | 0.90 | 0.72 | 0.33 |
| **Recognizing social norm violations** | **0.91** | **0.79** | **0.38** |

**Results for social commonsense categories**: Table 5 presents the performance of our model across different broad social commonsense categories, evaluated using BERTScore (F1), semantic similarity (SeSS), and ROUGE-L. Notably, for all four categories, the results are very similar showing the robustness of the design of **MemeSense**. The model achieves the highest scores in *recognizing social norm violations* (BERTScore: 0.91, SeSS: 0.79, ROUGE-L: 0.38), suggesting strong alignment with human references in identifying and intervening in socially inappropriate memes containing themes such as *vulgarity*, *sexual content* etc. For the other three categories also the results are quite close in terms of all three metrics (BERTScore: 0.90/0.91, SeSS: 0.72-0.78, ROUGE-L: 0.33-0.37).

## 8   Discussion

**Error analysis**   To better analyze the limitations of **MemeSense**, we conduct a detailed error analysis by examining its predictions and identifying cases where erroneous classifications occur. We categorize the errors into two types:
(1) *False negative* (Category 1 error): Instances where the meme is actually harmful and should be flagged as unsafe, but **MemeSense** incorrectly predicts it as safe for posting. Since non-harmful memes are absent from the dataset, the concept of false positives does not arise in our evaluation.
(2) *Improper reasoning* (Category 2 error): Cases where the model correctly identifies the meme as unsafe but provides incorrect or inadequate reasoning for its decision.
Among 51 such instances in our dataset, **MemeSense** exhibits Category 1 errors in 6 cases. Notably, in 5 out of these 6 cases, the commonsense parameter generation stage fails to accurately infer the harmful category, leading to incorrect classification. A specific example of this failure is observed when the model incorrectly identifies *cultural sensitivity* as the primary harmful category for a meme that is actually *vulgar*, ultimately leading to its misclassification as safe for posting.
Further, we identify one instance of Category 2 error, where the model predicts the meme as unsafe but fails to provide a coherent justification. This error arises due to improper reasoning during the commonsense parameter generation stage, which affects the interpretability and reliability of the model's intervention. A more detailed breakdown of the failure-cases is demonstrated in the Appendix E.

**Ablation studies**   In the error analysis, we observed the major prediction error appeared due to the incorrect generation of commonsense parameters. Hence we investigate, how much the final inference is dependent on the generated commonsense parameters. To achieve this, we obtain the inference from our approach without providing commonsense information to the model. Using only the input image and its corresponding description, we attempt to infer the intervention from our approach using the best method (**MemeSense** (combined)). The combined model is trained using the commonsense information. However, during the inference we are not providing the commonsense, removing the requirement of commonsense generation module during inference. We observe a maximum decline in semantic similarity score of 4%

without commonsense information. In addition, we observe that the interventions are more descriptive, which is reflected in the increase of the *readability* score.

**Effect of coefficient** $\alpha$   To understand the effect of coefficient $\alpha$ in the Equation 1, we conduct an ablation by setting $\alpha_i = 1$ (non-trainable), thereby isolating the effect of CSV. This resulted in a consistent performace drop accross all dataset. BERTScore decreased to 0.87 (4%) for for memes with and without text,

Table 6: Result for intervention generation for different test sets without coefficient $\alpha$.

| Test set | BERTScore (F1) | SeSS | Readability | ROUGE-L (Avg) | BLEU (Avg) |
|---|---|---|---|---|---|
| Memes without text | 0.87(-.04) | 0.61(-.1) | 41.56(-2.51) | 0.22(-.13) | 0.03(-.05) |
| Memes with text | 0.87(-.04) | 0.66(-.12) | 41.21(-2.17) | 0.25(-.11) | 0.03(-.05) |
| ICMM | 0.82(-.05) | 0.21(-.1) | 43.33(-2.24) | 0.07(-.04) | 0.006(-.002) |

and BERTScore reduced by 5% for ICMM dataset. Full result is shown in Table 6. These results suggest that removing the coefficient $\alpha$ leads to a notable decline in both semantic and surface-level quality of the generated interventions. $\alpha$ plays a crucial role in adaptively regulating commonsense infusion while generating intervention.

**MemeSense sensitivity analysis**   In addition to the ablation studies presented in Table 7, we conduct a sensitivity analysis to assess the impact of variations in the commonsense information provided to the model. Specifically, we evaluate how **MemeSense** (combined) performs when supplied with randomly selected commonsense knowledge during inference.

Table 7: Result for intervention generation for different test sets without using the commonsense parameters.

| Test set | BERTScore (F1) | SeSS | Readability | ROUGE-L (Avg) | BLEU-4 (Avg) |
|---|---|---|---|---|---|
| Memes without text | 0.89(-.02) | 0.68(-.03) | 51.02(+6.95) | 0.31(-.04) | 0.05(-.01) |
| Memes with text | 0.9(-.01) | 0.74(-.04) | 47.79(+4.41) | 0.32(-.04) | 0.06(-.02) |
| ICMM | 0.85(-.02) | 0.27(-.04) | 54.19(+8.62) | 0.10(-.01) | 0.007(-.001) |

This experiment aims to understand the model's sensitivity to incorrect or unrelated commonsense attributes.

As shown in Table 8, we observe a noticeable decline in performance across key metrics when randomly selected commonsense information is used. In particular, the semantic similarity score decreases by approximately 9%, indicating that misattributed commonsense knowledge can significantly affect the model's final outcome. The decline is also reflected in BERTScore, ROUGE-L, and BLEU, demonstrating the reliance of **MemeSense** on relevant commonsense reasoning for effective intervention generation. Interestingly, readability exhibits a slight improvement for memes with text, which could be attributed to the increased linguistic diversity introduced by the random commonsense selection. These findings highlight the importance of precise commonsense attribution in ensuring robust and reliable meme interpretation. We present a case study in Appendix D, where we examine the impact of commonsense reliability on the final intervention generation.

Table 8: Result for intervention generation for different test sets using randomly selected commonsense parameters.

| Test set | BERTScore (F1) | SeSS | Readability | ROUGE-L (Avg) | BLEU-4 (Avg) |
|---|---|---|---|---|---|
| Memes without text | 0.88(-.03) | 0.64(-.07) | 36.76(-7.31) | 0.27(-.08) | 0.05(-.03) |
| Memes with text | 0.89(-.02) | 0.69(-.09) | 46.36(+2.98) | 0.28(-.08) | 0.05(-.03) |
| ICMM | 0.85(-.02) | 0.27(-.04) | 34.07(-11.50) | 0.10(-.01) | 0.007(-.001) |

**Interpretability of cognitive shift vectors**   To assess the interpretability of CSVs and their correlation with commonsense parameters, we conduct two experiments as follows.

**Semantic consistency within commonsense parameters** We analyze whether CSV representations exhibit structured patterns within specific commonsense parameters. From the test set, we select five memes associated with a particular commonsense parameter and pass them through the **MemeSense** framework. We extract the hidden representations of the first generated token and compute the average pairwise Euclidean distance between these representations. In contrast, we repeat the process with five memes from different commonsense parameters. We observe that memes sharing a common parameter exhibit lower pairwise distances compared to those from mixed categories. For example, the average Euclidean distance among representations of memes labeled with "vulgarity" is **0.21**, whereas it increases to **0.28** when considering memes from multiple categories. This suggests that CSVs capture task-relevant semantic similarities.
**Correlation between commonsense parameters and representation similarity**: We investigate

whether hidden representations align with commonsense parameters that frequently co-occur. For instance, "vulgarity" often appears alongside "sexual content," while "stereotyping" commonly co-occurs with "Hate Speech." To analyze this, we select five memes from each of the top five most frequently co-occurring categories, process them through **MemeSense**, and compute the average pairwise Euclidean distances of the first generated token's representations. Our findings indicate a strong negative correlation ($\rho =$ **-0.67**) between category co-occurrence frequency and pairwise Euclidean distances. This suggests that conceptually related memes yield similar intervention representations, reinforcing the utility of CSVs.

These results suggest that CSVs effectively capture structured semantic patterns, supporting their role in task-relevant information distillation.

**Intervention quality measurement** To assess the quality of the generated intervention, we performed a quantitative and qualitative analysis as described below:

(1) *Measuring argument quality*: We aim to measure the argument characteristic of the generated response commonly used for measuring quality of online *counterspeech* Saha et al. (2024). We use a `roberta-base-uncased` model[14] finetuned on the argument dataset Stab et al. (2018). Given this model, we pass each generated intervention through the classifier to predict a confidence score, which would denote the argument quality. We obtain confidence scores of 0.67, 0.74, 0.79 for the memes without texts, memes with text, and the ICMM dataset respectively suggesting strong argument quality of the generated interventions.

(2) *Correlation with human judgments*: While we present most of our results with automatic metrics, it is important to understand if they correlate with human judgments. We took two metrics – BERTScore (F1) and ROUGE-L (Avg). For each metric, we randomly extract 25 samples from the prediction set. We present these to human annotators (researchers in this domain) and ask them to rate the quality of intervention from 1-5, 5 being the best and 1 being the worst. The Spearman's rank correlations between the human judgments (ordinal) and the automated metrics (continuous) are 0.58 and 0.49 respectively which indicates moderate to high correlation[15]. Given the subjective nature of the task, these results highlight a substantial consistency between automated metrics and human judgments, affirming their reliability.

(3) *Crowd based intervention evaluation*: We conduct a large-scale human evaluation on all 184 test memes (with and without overlaid text) using the Prolific platform, recruiting 81 high-approval-rate participants with prior experience and compensating them fairly. Each meme-intervention pair generated by **MemeSense** is independently evaluated by three raters, who choose one of three labels: (i) the intervention correctly addresses the harmfulness, (ii) the intervention is correct but the justification is incorrect, or (iii) both are incorrect. To ensure quality, we provide detailed task instructions, category definitions, examples, and a disclaimer about potentially harmful content. We observe fair inter-annotator agreement (Fleiss' $\kappa = 0.52$), which is reasonable given the subjectivity involved. Importantly, in 84% of cases, at least two annotators agree that the intervention correctly addresses the meme's harmfulness, supporting the practical effectiveness of our method.

**Generalizability across cultural and linguistic settings** Our current framework primarily targets memes containing English text or no text. To enhance diversity and evaluate broader applicability, we additionally include the **ICMM** dataset, which features numerous code-mixed examples (Hindi words in Latin script interspersed with English phrases). To assess cross-cultural and cross-lingual robustness, we further introduce two datasets: (1) a Hindi meme dataset Dubey et al. (2025) comprising memes with overlaid text exclusively in Devanagari script, and (2) the **Cm-Off-Meme** dataset Kumari et al. (2024) containing Hindi-English code-mixed memes. We annotate 50

Table 9: Comparative results of **MemeSense** using other models.

| Used model | Method | BERTScore (F1) | SeSS | Rouge-L (Avg) |
|---|---|---|---|---|
| | | **Memes without text** | | |
| Idefics-9B | MemeSense (*random ICL*) | 0.89 | 0.69 | 0.31 |
| | MemeSense (*combined ICL*) | 0.9 | 0.71 | 0.34 |
| OpenFlamingo-9B | MemeSense (*random ICL*) | 0.88 | 0.67 | 0.29 |
| | MemeSense (*combined ICL*) | 0.9 | 0.7 | 0.32 |
| | | **Memes with text** | | |
| Idefics-9B | MemeSense (*random ICL*) | 0.9 | 0.75 | 0.33 |
| | MemeSense (*combined ICL*) | 0.91 | 0.77 | 0.36 |
| OpenFlamingo-9B | MemeSense (*random ICL*) | 0.89 | 0.74 | 0.32 |
| | MemeSense (*combined ICL*) | 0.91 | 0.77 | 0.35 |
| | | **ICMM data** | | |
| Idefics-9B | MemeSense (*random ICL*) | 0.85 | 0.27 | 0.1 |
| | MemeSense (*combined ICL*) | 0.86 | 0.3 | 0.1 |
| OpenFlamingo-9B | MemeSense (*random ICL*) | 0.85 | 0.26 | 0.09 |
| | MemeSense (*combined ICL*) | 0.85 | 0.29 | 0.1 |

---

[14]https://huggingface.co/chkla/roberta-argument
[15]https://datatab.net/tutorial/spearman-correlation

examples from each dataset, using GPT-4o for initial commonsense generation followed by manual correction by two native Hindi speakers to ensure linguistic and cultural fidelity.

Using our fine-tuned `llava-1.6-mistral-7b-hf`, we first generate commonsense parameters. As expected, performance drops considerably for Devanagari-text memes due to the model's English-centric pretraining, whereas for the code-mixed Hinglish dataset, results improve substantially with only occasional failures arising from partial misinterpretation of Hindi tokens. The BERTScore of the Cognitive Shift Vector (CSV)-based interventions for the two datasets are **0.48** and **0.86**, respectively.

To further investigate whether multilingual vision-language models enhance intervention quality, we fine-tune `Llama-3.2-11B-Vision-Instruct` on our training instances for commonsense generation (Stage I). This yields significantly higher BERTScores of **0.85** and **0.89** for the two datasets, demonstrating that our framework readily adapts to multilingual settings with suitable backbone architectures. *Note that the CSV module is not trained in either case.*

**Runtime analysis**  Since our framework uses multiple stages to generate the final intervention, it is crucial to analyze computational efficiency of the framework. We compare the inference time of our approach with the k-shot LLM based approach on the ICMM dataset. Since, fine-tuning (stage I) and training cognitive shift vectors (stage III) are one time processes, it does not affect overall inference time. However, if we keep increasing the number of in-context examples in simple k-shot prompting, the computational cost as well as the inference time significantly increases. For instance, inference from 4-shot ICL will take 5.4x time compared to CSV, whereas inference from 8-shot ICL will take 9.1x time compared to CSV. However, inference from CSV will take only 1.2x time compared to standard zero-shot prompting. But the performance of zero-shot prompting is significantly poor (See the Table 2). For further understanding the trade-off between training + inference time of CSV compared to the k-shot prompting, we showcase the total time taken to infer from ICMM dataset in Table 10.

Table 10: Total runtime comparison.

| Method | Total Time |
|---|---|
| 0-shot (ICL) | 24.6 Min |
| 1-shot (ICL) | 57.45 Min |
| 2-shot (ICL) | 92 Min |
| 4-shot (ICL) | 160.8 Min |
| 8-shot (ICL) | 269.2 Min |
| MemeSense (Training + Inference) | **111.5 Min** (82 Min + 29.5 Min) |

**Use of alternative LLMs**  In the Table 9, we show the comparative results of MemeSense using different base LLMs (`Idfics-9B` and `OpenFlamingo-9B`). Here we use the annotated data mentioned in 4.1, and the retrieval of in-context exemplars mentioned in Section 4.2 to train the cognitive shift vectors (mentioned in Section 4.3) with these two base models. Then we perform the inference using trained cognitive shift vectors. We observe a similar pattern as earlier for these two LLMs. Moreover, `Idefics-9B` shows an overall superior performance compared to `OpenFlamingo-9B`.

## 9  Conclusion

In this work, we introduced MemeSense, a three-stage, adaptive in-context learning framework that integrates visual and textual cues with social commonsense knowledge for robust meme moderation. By combining compact latent representations, carefully retrieved in-context exemplars, and cognitive shift vectors, our approach captures subtle, implicitly harmful signals, *including memes without explicit text* that often evade traditional pipelines. Experiments on our curated dataset and the *ICMM* benchmark highlight MemeSense's superior performance in generating semantically aligned interventions, surpassing state-of-the-art baselines. We hope MemeSense inspires broader research in in-context learning toward fostering safer, more responsible online communities.

**Broader impact statement**

MemeSense introduces a socially grounded meme moderation approach using in-context learning with commonsense cues, enabling detection of subtle harms in text-light content. While promoting safer online spaces, it carries risks of overreach, cultural bias, and misuse. The authors aim to mitigate this through

transparent model release and advocate for culturally inclusive annotations and human oversight. Its real-world impact depends on responsible deployment and ethical safeguards.

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

## A  Prompts

The prompt for generating ground truth commonsense parameters and intervention using GPT-4o is represented in the Table 18. The prompts that we use for k-shot ICL based baselines are mentioned in the Table 19.

## B  Additional results

**Evaluating ICMM results under original MemeGuard configuration.** While Table 4 reports results using our adapted version of MemeGuard for fair comparison, we also evaluate its performance under the original setup to validate consistency. Specifically, we replace `idefics-2-8b-base` with `Flan-T5-Large`, the best-performing open-source LLM used in the original MemeGuard implementation. This yields modest improvements: BERTScore increases from 0.79 to 0.82, SeSS score goes from 0.18 to 0.25, ROUGE-L from 0.04 to 0.08, and BLEU-4 from 0.001 to 0.009. Despite these gains, the model's performance remains below that of **MemeSense** (BERTScore: 0.87, SeSS: 0.31, ROUGE-L: 0.11, BLEU-4: 0.009), reaffirming the advantage of our method. These findings highlight that language-centric LLMs like Flan-T5 can enhance zero-shot text generation, but still fall short in capturing nuanced multimodal harm without additional commonsense grounding.

Table 11: Result for the ICMM dataset. * indicates statistically significant improvement from MemeGuard and MemeMQA using *Mann-Whitney U test* with $p < 0.05$.

| Method | BERTScore (F1) | SeSS | Readability | ROUGE-L (Avg) | BLEU-4 (Avg) |
|---|---|---|---|---|---|
| Direct prompting | 0.8 | 0.15 | 67.02 | 0.03 | 0.001 |
| Direct prompting with commonsense | 0.8 | 0.14 | 52.34 | 0.03 | 0.004 |
| Random ICL | 0.82 | 0.16 | 19.63 | 0.09 | 0.005 |
| Image anchored ICL | 0.82 | 0.2 | 22.16 | 0.1 | 0.006 |
| Commonsense anchored ICL | 0.84 | 0.22 | 25.38 | 0.1 | 0.006 |
| MemeMQA | 0.85 | 0.24 | 54.45 | 0.1 | 0.007 |
| MemeGuard (Our settings) | 0.79 | 0.18 | 34.45 | 0.04 | 0.001 |
| MemeGuard (original) | 0.82 | 0.25 | 47.22 | 0.08 | 0.009 |
| MemeSense *(random ICL)* | 0.84 | 0.18 | 44.03 | 0.11 | 0.007 |
| MemeSense *(image anchored ICL)* | 0.85 | 0.25 | 42.79 | 0.1 | 0.007 |
| MemeSense *(commonsense anchored ICL)* | 0.86* | 0.27* | 42.22 | 0.11* | **0.009** |
| MemeSense *(combined)* | **0.87*** | **0.31*** | 45.57 | **0.11*** | 0.008 |

## C  Additional dataset details

We deliberately select only the harmful memes to build our MemeSense framework. Initially we collected a total of 785 memes from different online resources as mentioned in 5. We ask two undergraduate students to unanimously mark whether the memes are harmful or not. To maintain consistency, we provided them with a short annotation guideline, which included example images of both harmful and non-harmful memes (similar to Figure 3). More specifically, we ask them to mark a meme as harmful if it falls in the specified common sense category according to their judgments. This process resulted in 484 scrutinized harmful memes for our experiments. Since the memes that do not have embedded text in it, represents mostly sexually explicit items, our dataset contains a higher proportion of such memes (as reported in Table 1).

For the verification of the generated commonsense parameters and the intervention from the `GPT-4o`, we employ two expert reviewers to assess. We provide them with a short description of each commonsense category (Similar to the Commonsense parameters mentioned in the Table 18) along with one selected meme within each category for reference.They were allowed to update, delete, or add categories based on their judgment. Finally in 18 out of 484 cases they were required to correct the commonsense parameters and the corresponding interventions for a meme.

## D  Case Study: Impact of commonsense reliability on intervention generation

To further examine the sensitivity of **MemeSense** to the quality of commonsense input, we present a qualitative case study analyzing how variations in the generated commonsense parameters influence the

Table 12: Hyperparameters for **MemeGuard**.

| Hyperparameters | Task | Value |
|---|---|---|
| Temperature | Desc, Bias, Stereotype, Toxicity & Hate, Claim Generation | 0 |
| num_beams | Desc, Bias, Stereotype, Toxicity & Hate, Claim Generation | 1 |
| max_new_tokens | Desc, Bias, Stereotype, Toxicity & Hate, Claim Generation | 512 |
| Cosine Similarity Threshold | MKS Filtering | 0.2 |
| max_new_tokens | Intervention | 1024 |

Table 13: Prompt used for different tasks in the **MemeGuard** method.

| Task | Prompt |
|---|---|
| Description generation | Describe this meme in detail. |
| Social bias gen. | What is the societal bias that this meme is conveying? |
| Social stereotype gen. | What is the societal stereotype that this meme is conveying? |
| Toxicity and hate | What is the toxicity and hate that this meme is spreading? |
| Claim the meme is making | What are the claims that this meme is making? |
| Intervention Generation | This is a toxic meme with the description: ks1. The following text is written inside the meme: X. Rationale: Bias: ks2, Toxicity: ks3, Claims: ks4, and Stereotypes: ks5. Write an intervention for this meme based on all this knowledge. |

final intervention. This analysis builds upon the findings in Table 8, where we measured performance under randomly selected commonsense attributes.

Our observations reveal two consistent patterns:

1. **Robustness through partial accuracy**: In cases where at least one of the predicted commonsense parameters aligns with the ground truth, **MemeSense** often succeeds in generating a contextually appropriate intervention. This suggests that the model is capable of leveraging even partial commonsense grounding to orient the cognitive shift vector in a meaningful direction, thereby preserving semantic and ethical relevance in the intervention.

2. **Intervention disruption via semantically divergent commonsense**: When the predicted commonsense parameters are semantically distant or rarely co-occurring with the ground truth categories-e.g., substituting *Vulgarity* with *Cultural Sensitivity*-we observe a marked decline in intervention quality. In such cases, the model's attention appears to shift toward an unrelated ethical dimension, resulting in generic or misaligned interventions.

These findings suggest that while **MemeSense** exhibits a degree of resilience to noisy commonsense input, its performance is sensitive to the semantic proximity between the predicted and actual commonsense parameters. We include representative examples below to illustrate both robust and failure cases under random commonsense perturbation. We include representative examples in Table 14 to illustrate both robust and failure cases under random commonsense perturbation.

# E  Comprehensive failure case analysis

Table 15 summarizes representative failure cases from **MemeSense** where the model's commonsense analysis surfaced useful cues but the final intervention label was incorrect. Typical errors include (i) under-detecting sexual innuendo in visually ambiguous memes, (ii) overlooking cultural or religious sensitivity despite partial

Table 14: Case study illustrating examples where randomly provided commonsense either preserves or disrupts the quality of the generated intervention. The correct commonsense category is highlighted in green and the semantically divergent commonsense is highlighted in red.

| Case | Case 1: Intervention not affected by random commonsense | Case 2: Intervention affected by random commonsense |
|---|---|---|
| **Meme image** |  |  |
| **Provided commonsense to the model** | **Vulgarity:** The text includes inappropriate language and innuendos referencing male genitalia in a metaphorical context.

**Humor Appropriateness**: The humor is based on a sexual analogy that might be perceived as crude or offensive, especially given the literal image contrast. | **Cultural Sensitivity**: The meme references Mughal art and Pakistani identity, involving cultural and historical elements. It may be seen as misrepresentative or disrespectful.

**Humor Appropriateness**: The humor stems from anachronism - depicting a historical figure engaging in a modern sport-which some may find mocking or belittling cultural heritage. |
| **Predicted intervention** | The meme should not be posted publicly due to its vulgar language and inappropriate humor, which may offend or discomfort viewers. | The meme may be perceived as culturally insensitive due to its use of Mughal art and Pakistani identity in a humorous context. |
| **Actual intervention** | The meme is inappropriate for public posting due to its vulgar innuendo and potentially offensive humor. It may not be suitable for all audiences and could lead to misunderstandings or discomfort. | The meme should not be posted publicly as it includes suggestive sexual content that is inappropriate for a broad audience. The humor could be seen as offensive or in poor taste, possibly provoking negative reactions. |

hints in the reasoning, (iii) misjudging misogynistic framing as benign "humor", and (iv) over-weighting salient visuals (e.g., a crying face) while missing explicit vulgarity in text. These patterns suggest limits of our cognitive-shift and in-context exemplars when cues are subtle, multi-cultural, or context-entangled. We view these as actionable diagnostics: strengthening cross-modal evidence aggregation, bias-aware reasoning for sensitive domains, and targeted instruction for stereotype and innuendo detection should reduce such errors.

## F   Qualitative comparison with explainable SOTA methods

We conduct a qualitative comparison of MemeSense with two recent explainable meme classification frameworks – IntMeme Dubey et al. (2025) and U-CoT+ Hee & Lee (2025). IntMeme first extracts the overlaid caption and processes it through a frozen large multimodal model (LMM) to generate an interpretation. This interpretation, combined with visual features, is then used to train a classifier for binary hatefulness detection. Although IntMeme is not directly comparable due to its binary classification focus, we contrast its generated interpretations with the commonsense outputs of MemeSense. We find that IntMeme struggles to capture nuanced or implicit harms, especially in memes without textual overlays, and often produces partial or inadequate explanations even when text is present.

We also evaluate U-CoT+, which generates zero-shot explanations using multiple LLM calls and handcrafted task-specific prompts. While it offers brief decoupled explanations, its reliance on multiple inference stages and domain-specific tuning limits its real-time applicability. In contrast, MemeSense demonstrates greater scalability and adaptability, producing context-aware interventions efficiently – even for visually implicit or

Table 15: Representative failure cases from MemeSense intervention analysis.

| Meme | Predicted intervention | Ground truth | Analysis |
|---|---|---|---|
|  | The meme does not appear to be vulgar, harmful, or unethical. | Restrict due to sexual innuendo that may be unsuitable in some cultural contexts. | Generated commonsense noted potential cultural sensitivity (hand gesture misinterpretation), but the final prediction still marked it as non-harmful. Illustrates cases where cognitive-shift or in-context cues fail in isolation. |
|  | The meme does not appear to be vulgar or harmful. | Restrict due to sexually suggestive interpretation. | Commonsense flagged suggestive visual cues, yet visual ambiguity led to a non-harmful decision. |
|  | The meme does not appear to be vulgar or harmful. | Restrict due to sexually suggestive reading of the building's structure. | Commonsense hinted at inappropriate humor from unusual structure, but the final decision remained non-harmful. |
|  | Review due to possible mental-health impact. | Restrict for vulgar wording and disrespect toward subjects. | Prediction overlooks explicit vulgarity, over-weighting the visual context (e.g., "crying person"). |
|  | Review for potentially inappropriate humor in some cultural contexts. | Restrict for misogynistic framing and suggestive innuendo. | Commonsense misses misogyny: implies derogatory view of women as talkative/nagging, promoting sexism. |
|  | Restrict due to suggestive innuendo not suitable for all audiences. | Restrict for vulgar depiction and potential offense - especially given the religious context and humor about a real location. | Harmfulness is detected, but commonsense fails to capture religious sensitivity. |

text-free memes – through lightweight cognitive shift vector training. Example in Table 17 illustrate these comparative insights.

## G  Additional experimental settings

### G.1  Baselines

In Table 13 we demonstrate the different prompts used for the **MemeGuard** baseline. The hyperparameters for the experiments with this baseline are noted in Table 12.

## H  Representative examples of memes from each commonsense category

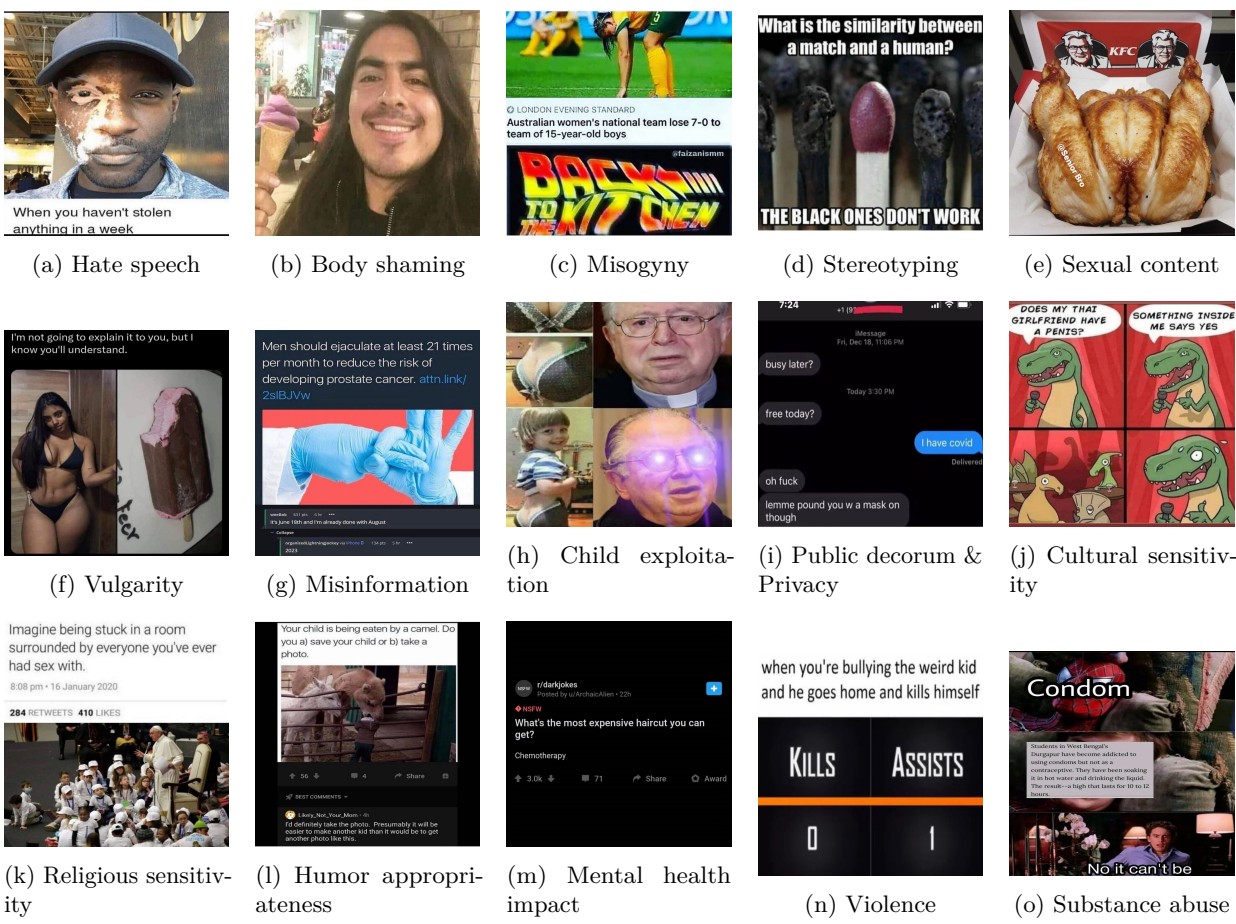

Figure 4: Representative examples of memes from each of the 15 commonsense harm categories.

Table 16: Generated intervention using different approaches for (1) a meme without text, (2) a meme with text, (3) a meme from ICMM data. Incorrect justifications are highlighted.

| | (1) | (2) | (3) |
|---|---|---|---|
| Direct prompting | The meme is a popular form of communication that has been around for centuries. It is a way to express ideas, opinions, and emotions through images and text. Memes can be funny, serious, or even offensive. They can be used to spread information, make a point, or just entertain. | The meme is a humorous image that depicts a woman pouring milk into a container. The text on the image reads, "Her: "Just make sure you don't finish inside of me." Me inside of her 2 minutes ago." The meme is a play on the phrase "just make sure you don't finish inside of me," which is often used to indicate that a person should use a condom during sex. The meme is humorous because it suggests that the woman is concerned about the man finishing inside of her, but the man is actually inside of her at the time. | The meme is a picture of a man with the caption "when the school shooter finally leaves and the hottest girl in class is laying dead on the floor". The meme is making fun of the school shooter and the girl who is dead. The meme is also making fun of the fact that the girl is dead. |
| Direct prompting (w. commonsense) | In the image, a person is sitting on a chair, holding a hot dog on a stick The hot dog is being cooked on a grill, and the person is enjoying the process There are two other hot dogs on the grill, one located closer to the person and the other further away In addition to the hot dogs, there are two marshmallows on the grill, one near the person and the other further away The overall atmosphere appears to be a fun and relaxed outdoor gathering. | The meme is inappropriate and violates community guidelines against explicit or adult content. It is not suitable for general audiences and may be offensive or uncomfortable to some viewers. | The meme is a screenshot of a man smiling and posing for the camera. The text on the image reads "When the school shooter finally leaves and the hottest girl in class is laying dead on the floor its free real estate". The combination of the text and image attempts to make a joke about a potentially tragic situation by using dark humor. The meme references a school shooting a highly sensitive and traumatic subject. Making jokes about such an event can be perceived as trivialising a grave and distressing experience, potentially causing harm to those affected. |
| Random ICL | Celebrating individuality and embracing diversity enhances understanding and acceptance among us all. | This meme is vulgar, inappropriate, and potentially harmful, as it promotes a misogynistic view of sexual consent. | The meme should not be posted publicly due to its insensitive and inappropriate content, which could offend or disturb certain audiences. |
| Image anchored ICL | This meme humorously highlights the playful creativity and shared sense of humor in relationships, encouraging laughter and bonding over shared jokes. | This meme is vulgar, inappropriate, and potentially harmful, as it promotes a misogynistic view of sexual consent. | The meme should not be posted publicly as it trivializes the trauma of school shootings and can be distressing to individuals who have experienced similar events. It is inappropriate to use humor in this context, as it can be perceived as insensitive and offensive. |
| Commonsense anchored ICL | The image features a man holding a hot dog on a stick, with a marshmallow on one end and a penis on the other This meme is disrespectful and inappropriate, as it makes light of a serious issue, such as erectile dysfunction, and uses it to create a humorous or entertaining situation. It is important to be aware of the impact of such memes and to promote respectful and inclusive communication. | The meme should not be posted publicly due to its vulgar and suggestive content, which could be offensive to many viewers. | The meme appears to be inappropriate for public posting due to its insensitive and offensive content. It trivializes the trauma of school shootings and could be distressing to individuals who have experienced similar events. The use of humor in this context may be perceived as insensitive and offensive, particularly considering the serious nature of school shootings. The subject matter could be perceived as a violation of public sensitivity and a discussion of serious topics in a public context. |
| MemeQA | The meme should be restricted due to its content and content. It is inappropriate for public posting and should not be shared publicly. | The meme should be restricted due to its content and inappropriate content. | The meme should not be posted publicly due to its potential to be offensive to some audiences. |
| MemeGuard | The meme is making a claim that the person is having a barbecue with a hot dog and marshmallows, but instead of a traditional hot dog, there is a penis-shaped hot dog. The meme is likely meant to be humorous and playful, as it takes a common outdoor activity like grilling and adds a provocative twist to it. | The meme is harmful because it reinforces negative stereotypes about women and their sexuality. It suggests that women are concerned about pregnancy and STIs, and that they are not interested in sexual pleasure. It also suggests that men are not concerned about these issues and are only interested in their own pleasure. This is harmful because it perpetuates the idea that women are not equal to men in terms of sexuality and that they are not entitled to sexual pleasure. | This meme contains highly inappropriate and offensive content, trivializing school shootings and making an unethical implication about victims. This kind of meme should be critically examined and condemned rather than shared |
| MemeSense (Random ICL) | The meme should be restricted from public posting due to its vulgar and suggestive nature, which could be perceived as offensive or inappropriate. | The meme should not be posted publicly due to its vulgarity, sexual content, and inappropriate humor. | The meme should not be posted publicly due to its insensitive and potentially traumatizing content. It trivializes a serious issue and may cause distress to individuals who have experienced similar events. |
| MemeSense (Image anchored ICL) | The meme should not be posted publicly due to its vulgar and suggestive nature, which could be perceived as offensive or inappropriate. | The meme should not be posted publicly due to its vulgar, sexual, and inappropriate nature. It could be perceived as offensive and disrespectful to many audiences. | The meme should not be posted publicly due to its insensitive and inappropriate content. It trivializes a serious and sensitive topic, which can be distressing to individuals who have experienced school shootings or related trauma. |
| MemeSense (Commonsense anchored ICL) | The meme should be restricted from public posting due to its vulgar and suggestive content, which could be perceived as offensive or inappropriate. | The meme should not be posted publicly due to its vulgar and suggestive content, which could be offensive to many viewers. | The meme should not be posted publicly due to its insensitive and potentially traumatizing content. It trivializes a serious issue and may cause distress to individuals who have experienced similar events. |
| MemeSense (Combined) | The meme should be restricted from public posting due to its vulgar and suggestive content, which could be perceived as inappropriate. | The meme should not be posted publicly due to its vulgar and suggestive content, which could be offensive to many audiences. It also uses humor that could be perceived as belittling and inappropriate. | The meme should not be posted publicly due to its insensitive and potentially traumatic references to school shootings. It trivializes a serious issue and may cause distress to individuals who have experienced similar events. |
| Ground truth annotation | The meme should be restricted in certain contexts due to its potentially suggestive content, which might be perceived as vulgar or inappropriate, particularly in public or professional settings. | The meme should not be posted publicly due to its vulgar and sexually suggestive nature, which may be offensive and inappropriate for general audiences. | Joking about tragic events such as school shootings and the loss of innocent lives is insensitive and inappropriate. We should be mindful of the impact our words can have on others and strive to promote empathy and understanding. |

Table 17: Qualitative comparison of explanations/interventions generated by IntMeme, U-CoT+, and MemeSense. All models are evaluated on the same meme input.

| Meme |  |
| --- | --- |
| **Explanation by** IntMeme | The meme shows a black man sitting next to a UPS truck, with his pants down around his ankles. The text "UPS man's package" is overlaid on the image, indicating that it might be a sexual joke or a reference to the UPS employee having to deliver a package with an unusual or unexpected item. The presence of this meme could be construed as a form of racial bias and stereotyping, as it portrays black men as being less responsible, less professional, and possibly even less respectful. The meme may also be seen as objectifying and sexualizing black men, reinforcing harmful stereotypes about the African-American community. |
| **Explanation by** U-CoT+ | *Step-by-step analysis:* The image shows two individuals, one in a uniform-like shirt (possibly a delivery driver, given the partially visible "UPS" logo on the vehicle) and another person, seemingly in close physical contact inside the open rear door of a dark-colored van. The image is blurry, and the subjects' actions are suggestive of an intimate or sexual encounter. Alignment with Guidelines: Sexualization of Women: The depiction of an intimate act in a public/work-related setting, particularly focusing on the apparent female subject being treated as an object for sexual desire, aligns with the guideline on Sexualization of Women and potentially Objectification of Women. While the full context (a caption) is missing, the visual content itself focuses on a sexual encounter that can be interpreted as treating one person (the woman, given her typical attire compared to the other person) as an object to satisfy sexual desire. |
| **Commonsense generated by** MemeSense | The posture and context imply a suggestive or intimate scenario between the two individuals. **Public Decorum & Privacy:** The image captures a private moment in a public or professional setting, potentially without consent, thereby raising ethical and cultural concerns. |

Table 18: Prompt to generate the ground-truth commonsense and interventions.

**Prompt used for obtaining commonsense parameters**

As an AI assistant for social media content moderation, your role is to analyze and prevent the posting of harmful, offensive, hateful, vulgar, misogynistic, or otherwise inappropriate content on public platforms.

Task: Carefully analyze and interpret the given image, paying close attention to both visual elements and any text present.

1. Meme Description: Provide a detailed description of the meme, including any text, imagery, or symbolic references.

2. Commonsense Analysis:
* Assess whether the meme has the potential to be perceived as vulgar, harmful, or unethical.
* Identify and categorize up to 5 specific commonsense parameters from the predefined list below.
* Each selected category should be explicitly justified with a brief explanation.

Predefined Commonsense Categories (Select Up to 5):
* Vulgarity: Use of obscene language, inappropriate tone, or implied innuendos.
* Violence: Depictions of harm, coercion, or references to physical aggression.
* Sexual Content: Explicit or suggestive imagery, innuendos, or nudity.
* Hate Speech: Insulting, harassing, or offensive content targeting individuals or groups.
* Body Shaming: Objectification, fat-shaming, or insensitive references to disabilities.
* Cultural Sensitivity: Stereotypes, appropriation, or misrepresentation of cultural elements.
* Mental Health Impact: Content that manipulates emotions, triggers trauma, or exploits grief.
* Humor Appropriateness: Use of humor in a way that mocks, belittles, or distorts serious topics.
* Misinformation: Spreading misleading narratives, false claims, or deceptive dialogue.
* Substance Abuse: References to drugs, alcohol, or promoting substance use.
* Child Exploitation: Content that inappropriately depicts or exploits children.
* Public Decorum & Privacy: Violations of privacy, public sensitivity, or personal reputation concerns.
* Stereotyping: Generalizations that reinforce racial, gender, or societal biases.
* Misogyny: Content promoting gender-based discrimination, sexism, or demeaning women.
* Religious Sensitivity: Content that disrespects religious beliefs, symbols, or historical context.

3. Intervention Recommendation:
* If the meme is deemed inappropriate, justify why it should not be posted publicly.
* If the content is safe, confirm its appropriateness.

Response Format:

Meme Description:
<Provide a detailed description of the meme, including text and images.>

Commonsense Analysis:
- **[Category Name]**: [Justification]
- **[Category Name]**: [Justification]
- **[Category Name]**: [Justification]

Intervention Recommendation:
<Explain whether the meme should be restricted and why.>

Table 19: Prompt used for different methods.

| Method | Prompt |
|---|---|
| **Direct prompting** | \<Meme\> Analyze the meme thoroughly, considering its message, symbolism, cultural references, and possible interpretations. Identify any implicit or explicit harm, misinformation, or reinforcement of negative stereotypes. Based on this analysis, generate strategic interventions to discourage the spread or creation of such content. These interventions should be precise, contextually relevant, and designed to effectively deter users from posting similar memes. They may include subtle deterrents, educational messaging, content reformulation, or alternative framing that neutralizes harmful intent. Ensure responses are concise, non-repetitive, and avoid redundant explanations. Ensure the response should not exceed 50 words. |
| **Direct prompting with commonsense** | \<meme\> Analyze the meme thoroughly, considering its message, symbolism, cultural references, and possible interpretations. Identify any implicit or explicit harm, misinformation, or reinforcement of negative stereotypes. Based on this analysis, generate strategic interventions to discourage the spread or creation of such content. These interventions should be precise, contextually relevant, and designed to effectively deter users from posting similar memes. They may include subtle deterrents, educational messaging, content reformulation, or alternative framing that neutralizes harmful intent. Ensure responses are concise, non-repetitive, and avoid redundant explanations. The common sense parameters associated with the meme is as follows: {*common_sense*} Ensure the response should not exceed 50 words. |
| **MemeMQA** | \<meme\>Analyze this meme and generate a caption that enhances its humor, sarcasm, or irony. Do not filter for offensiveness-prioritize humor, satire, or dark humor as needed. The caption should be punchy, relatable, and aligned with the meme's tone. |
| **ICL** | \<meme\> As an AI assistant tasked with social media content moderation, your role is to prevent harmful, offensive, hateful, vulgar, misogynistic, or unethical content from being posted on public platforms.\n \n Your Task: A toxic meme has the description below along with few commonsense parameters which assess whether the meme has the potential to be perceived as vulgar, harmful, or unethical. Write an intervention for the this toxic meme to discourage user posting such memes based on provided knowledge. {*commonsense_parameters*} \n \n {*examples*} |

