# OpenReview forum: "MemeSense: An Adaptive In-Context Framework for Social Commonsense Driven Meme Moderation"
_TMLR — Accepted by TMLR_

### Review · Reviewer_bQ8p · 2025-07-19

**Summary Of Contributions:**

### 1. Paper Summary

This paper proposes a meme content moderation framework driven by social common sense, MemeSense. This method combines visual and text understanding through a three-stage process (common sense parameter generation, in-context example retrieval, and cognitive shift vector learning) to improve the ability to identify and intervene in harmful memes with implicit or no text. The main contributions of the paper include: (1) designing an adaptive intervention generation process that combines social common sense prompts and dynamic in-context examples; (2) constructing a meme dataset covering 15 types of social common sense harm; (3) achieving superiority over state-of-the-art methods on benchmark datasets, with significant improvements in BERTScore.

### 2. Strengths

- MemeSense first introduced the cognitive shift vector into the meme moderation scenario, combined with social common sense, in-context learning, and multimodal modeling, overcoming the problems of traditional models' strong dependence on text and weak ability to identify hidden harm.
- Comprehensive evaluation, covering multi-dimensional automatic metrics such as BERTScore, ROUGE-L, BLEU, and supplemented by human consistency analysis.
- Significant results. On both the self-built dataset and the ICMM dataset, MemeSense's overall (such as BERTScore and SeSS) have been greatly improved, and the statistical significance of the improvement has been verified through significance tests.

### 3. Weaknesses

- There is a jump between evidence and conclusion. There is a lack of empirical or case analysis on the generalization ability of the model in actual complex scenarios (such as memes in multilingual and cultural environments). The article claims to have "stronger adaptability" but has only been verified on a limited number of English social media memes.
- Ablation and error analysis are not in-depth enough. Although the author has ablated the contribution of cognitive shift vector and in-context exemplar, there is a lack of qualitative analysis of the model performance in their respective failure scenarios, especially for misclassification cases; there is insufficient reasoning process tracking and failure cause location.
- The granularity of manual evaluation is coarse. The sample size of manual evaluation in the paper is limited (n=25), and only involves the rank correlation between BERTScore/ROUGE and subjective evaluation. No qualitative analysis of typical failure cases is conducted, and there is also a lack of practical application feedback at the user level (moderator/workflow).
- Dataset annotation and statistical bias risk. The self-built dataset is double-annotated by a small number of undergraduates + experts. Although Cohen's kappa is high, it does not fully explain the consistency and diversity of extreme/ambiguous meme types. The definition of the "potentially harmful but no text" category is highly subjective, and potential bias is not evaluated.
- Model interpretability is limited. Although there is a vector space clustering analysis of the cognitive shift vector, the interpretability of a specific inference process is still insufficient. There is a lack of transparent decision logs or link examples for actual moderator tracing or intervention recommendation reasons.

**Audience:**

Yes

**Audience Explanation:**

This paper explores an important challenge at the intersection of multimodal learning, content moderation, and social commonsense reasoning — topics that are highly relevant to the TMLR audience. The paper proposes the MemeSense framework, introduces new technical methods, and releases a new dataset that will be of interest to researchers working on visual language models, harmful content detection, and AI ethics.

**Broader Impact Concerns:**

While this method improves the meme moderation, it also brings risks such as "over-censorship" and cultural discrimination/accidental injury, especially in the vaguely defined areas of social common sense (such as humor and cultural symbols), which are prone to controversy. If certain mainstream culture/values are solidified during model training or reasoning, it may cause improper suppression of the freedom of expression of minority groups. Therefore, it is recommended that the author further analyze the potential bias risks of the model in multicultural scenarios and clearly disclose the uncontrollable scenarios and consequences of misjudgment of the model.

**Claims And Evidence:**

No

**Claims Explanation:**

Some claims—for example, generalization to multiple languages or cultural diversity, and applicability to real-world workflows—are not fully substantiated. The paper also provides a relatively limited analysis of model failures and annotation biases. Thus, the evidence for the paper’s technical contributions is compelling, but the claims about generalization and practical deployability are less robust.

**Requested Changes:**

1. It is recommended to introduce cross-cultural and cross-language memes (such as non-English corpus, different religious/regional backgrounds) for supplementary experiments to verify the practical applicability and robustness of the method.
2. The reasoning chain analysis of the failure case should be supplemented, and specific examples of memes when the cognitive shift vector/in-context example fails alone should be given to help understand the model boundary.
3. It is recommended to expand the sample size of human evaluation and invite actual moderators to conduct case evaluation to supplement the model usability and user perception indicators.
4. Supplement the analysis of data annotation consistency and bias. For the highly subjective textless meme harm category, it is recommended to supplement expert diversity annotation, boundary discussion, and discussion of potential cultural bias risks.
5. It is recommended to add visualization or decision tracing tools for the model reasoning chain to facilitate tracking and auditing the specific reasons for AI moderation in actual application scenarios.

---

> ### Author Response · Authors · 2025-08-17
> **We express our gratitude to reviewer bQ8p for insightful and constructive evaluation**
>
> **RC1:**  It is recommended to introduce cross-cultural and cross-language...
>
> **Answer:** Thank you for the suggestion. Our current method primarily targets memes containing **English text or no text**. To maintain diversity, we also include the ICMM dataset, which contains many code-mixed examples (Hindi words in Latin script mixed with English phrases). **As suggested by you**, to further evaluate the cross-cultural and cross-language robustness of our framework, we introduced two additional datasets:
> - (1) Hindi memes [1] containing overlaid text solely in Devanagari script, and
> - (2) Cm-off-meme dataset [2] containing Hindi–English code-mixed dataset. We rigorously annotated 50 examples from each set, using GPT-4o for initial generation followed by manual correction by two native Hindi speakers to ensure linguistic and cultural accuracy.
>
> First we use our fine-tuned *llava-1.6-mistral-7b-hf* to generate the common-sense parameters. However, we observe that **it struggles to address the Hindi scripts in the Hindi memes (which is expected given its English-centric pretraining)**. However, in the case of the Hinglish dataset it gives significantly better outcomes with few failure cases due to the misinterpretation of the Hindi texts. The BERTScore for the CSV generated interventions for two datasets were: 0.48 and 0.86 respectively.
>
> To further check whether multi-lingual VLM can enhance the intervention, we use *Llama-3.2-11B-Vision-Instruct* to fine-tune using our training instances for the commonsense generation (Stage I). We observed a significantly higher BERTScore for these two datasets - 0.85, 0.89 respectively, underscoring that **our framework can readily adapt to multilingual settings with the right backbone choice**.
>
> Note that, we do not train the CSV in either of the cases. We will add the detailed discussion in a **separate supplementary section**.
>
> [1] Dubey, Kriti, et al. "Multimodal Detection of Offensive Content in Hindi Memes." ACM Transactions on Asian and Low-Resource Language Information Processing.
>
> [2] Kumari, Gitanjali, et al. "Cm-off-meme: Code-mixed hindi-english offensive meme detection with multi-task learning by leveraging contextual knowledge." Proceedings of the 2024 Joint International Conference on Computational Linguistics, Language Resources and Evaluation (LREC-COLING 2024). 2024.
>
> ---
>
> **RC2:** The reasoning chain analysis of the failure case...
>
> **Answer:**  We appreciate your suggestion regarding supplementing the reasoning chain analysis of failure cases. In the current submission,
> - **we have included an illustrative example of method robustness in Appendix C**,
> - along with **reasoning chains for each type of baseline method and our method across the three datasets in Table 14 of the Appendix**.
> - **Section 8 also presents a comprehensive error analysis.**
>
> To further address your concern, **we will expand this analysis by adding a detailed breakdown of the six specific failure cases (as mentioned in Section 8) we have identified**. For each case, we will present the exact generated commonsense outputs and the corresponding interventions in a tabular format similar to **Table 13 in the Appendix**.
>
> For example, in one such case, **the generated commonsense highlighted a potential cultural sensitivity (“hand gesture could be misinterpreted as offensive in some contexts”)**, but the final prediction **incorrectly indicated that the meme did not appear to be harmful**. Including these additional examples will make it easier to understand scenarios where the cognitive shift vector or in-context example fails in isolation, which can provide a better insight into the boundaries and limitations of our method. Other examples are added in this anonymous link: https://anonymous.4open.science/r/MemeSense/harmful_meme_results%20-%20Error%20Cases.csv. We will update these results in the Appendix.
>
> ---
>
> **Please see the next section for more details (5K limit)**

---

> ### Author Response · Authors · 2025-08-17
> **Continuation of previous comment**
>
> **RC3:** It is recommended to expand the sample size...
>
> **Answer:** We acknowledge the reviewer’s suggestion and agree that expanding human evaluation can strengthen our study. **In addition to the small-scale evaluation reported in Section 8, we have now conducted a large-scale human evaluation on all 184 test instances (including both memes with and without overlaid text) using the Prolific [https://www.prolific.com/] platform**. To ensure annotation quality, **we recruited only high-approval-rate annotators with prior experience**, compensating them at **£18/hour**. We provided detailed task instructions, category definitions, representative examples, and a disclaimer about potentially harmful content. **A total of 81 participants contributed, with each meme evaluated by three independent raters to reduce individual bias**.
> For each meme–intervention pair generated by our MemeSense method, raters selected one of three labels:
> - Intervention correctly addresses the harmfulness in the meme,
> - Intervention is correct but justification is incorrect,
> - Both intervention and justification are incorrect.
>
> We observed a **fair inter-annotator agreement (Fleiss’ κ = 0.52 across the three labels) which is pretty good given the subjective nature of the task**. Notably, **in 84% of cases, at least two annotators agreed that the intervention correctly addressed the harmfulness**. These results demonstrate the practical effectiveness of our approach in mitigating harmfulness in memes. We will add this expanded human evaluation, along with detailed instructions, as a new subsection in Section 8.
>
> ---
>
> **RC4:** Supplement the analysis of data annotation...
>
> **Answer:** Thank you for your thoughtful comment. We fully agree that subjectivity and cultural context can significantly influence harm perception, particularly in textless memes where interpretation often relies on visual metaphors or culturally embedded cues.
>
> To explore this, **we conducted an additional consistency check across our human evaluations**. In 8 specific instances, all three raters provided divergent judgments. Upon manual inspection of these memes **(examples shown in the anonymous link: [https://anonymous.4open.science/r/MemeSense/Prolific_cultural_bias.csv](https://anonymous.4open.science/r/MemeSense/Prolific_cultural_bias.csv), including the sandal illusion, the heart-to-phallus shirt distortion, and context-specific food humor)**, we observed that each requires nuanced cultural interpretation—whether it’s symbolic vulgarity, implicit visual misdirection, or culturally resonant puns. **These findings reinforce the idea that culturally grounded background knowledge is often essential for disambiguating such cases**.
>
> We will explicitly highlight this challenge in the final version, along with examples, and clarify that the framework's error cases often lie at the boundary of cultural subjectivity—warranting more diverse, cross-cultural annotation in future work.
>
> ---
>
> **RC5:** It is recommended to add visualization...
>
> **Answer:** We appreciate the suggestion regarding visualization and decision tracing tools. **At present, we carefully analyze the model’s reasoning process through manual inspection of error cases, as presented in Section 8 and the Appendix**. This allows us to meaningfully identify corner cases and interpret where the method succeeds or fails. In future extensions, a promising direction would be to integrate our framework with a reasoning-focused model, which could enable automatic tracing of reasoning chains and their visualization in a way that supports real-world auditing needs.
>
> ---
>
> **NOTE:**
>
> In response to the question **“Would at least some individuals in TMLR's audience be interested in knowing the findings of this paper?”** you selected **No**. However, in your accompanying justification, you wrote:
>
> “This paper explores an important challenge at the intersection of multimodal learning, content moderation, and social commonsense reasoning — topics that are highly relevant to the TMLR audience. The paper proposes the MemeSense framework, introduces new technical methods, and releases a new dataset that will be of interest to researchers working on visual language models, harmful content detection, and AI ethics.”
>
> This justification clearly aligns with a **Yes** response. We believe the **No** selection may have been made in error, and we kindly request you to revise it accordingly.

---

> > ### Comment · Reviewer_bQ8p · 2025-08-17
> >
> > Thank you. I have changed No into Yes. It was indeed a mistake.

---

> > > ### Author Response · Authors · 2025-08-19
> > > **Kindly respond**
> > >
> > > Dear Reviewer bQ8p,
> > >
> > > As the author response period draws to a close, we hope that we have adequately addressed your concerns. We would greatly appreciate it if you could kindly acknowledge our response. Please let us know if there are any remaining points you would like us to clarify or expand upon.
> > >
> > > Thank you once again for your thoughtful and constructive feedback.
> > >
> > > Best regards,
> > >
> > > The Authors

---

> > > > ### Author Response · Authors · 2025-08-24
> > > > **Kindly respond**
> > > >
> > > > Dear Reviewer bQ8p,
> > > >
> > > > Please respond. We have tried to address your feedbacks.
> > > >
> > > > Best
> > > >
> > > > Authors

---

### Review · Reviewer_Qozu · 2025-08-01

**Summary Of Contributions:**

The study introduces a framework designed to detect harmful memes, including both image-only and image-with-text memes. The method demonstrates promising performance on benchmarks curated by the team, which comprise existing benchmarks as well as additional data collected from the internet.

The framework is organized into three primary stages: generating commonsense parameters for an image, selecting in-context examples(similar visually/semantically via ANN search), and learning cognitive shift vectors for the dataset. The commonsense parameters are based on a predefined taxonomy of harms derived from previous research in the field. The technique for computing  cognitive shift vectors is also from past work in the field. But the authors stack these components together for harmful meme detection which is novel. ICV also allows for the distillation of task-relevant information, reducing the need for multiple in-context examples during inference, which in turn lowers inference costs and time.

The paper should specify that it claims SOTA in post-hoc techniques(inference based that don't require training a model) for meme detection and only for the specific model sizes studied, rather than asserting SOTA across all models/methods. Including experiments with models from the Memeguard paper would be beneficial, as the reported numbers appear to be inconsistent. The writing could also be improved to reduce back-and-forth between sections. Providing more information about the methodology and dataset, as outlined below, would make this more reproducible.

**Audience:**

Yes

**Audience Explanation:**

Yes, the paper tackles the complex issue of moderating "nuanced harmful memes," which often mask their intent through humor, or cultural symbolism and frequently evade conventional moderation systems, especially those relying on explicit text. It's interesting to see how far one can push a model without finetuning to perform well at this task by just performing test time intervention that performs better than direct zero shot prompting.

**Broader Impact Concerns:**

No major concerns. However, since part of their dataset includes content sourced from the internet, particularly sensitive material like phallic memes, it would be helpful for the authors to clarify whether they obtained consent or if the data was available under a free license.

**Claims And Evidence:**

Yes

**Claims Explanation:**

The claims made in the submission are to a large extent accurate but there are some areas that could benefit from further clarification and additional data to strengthen the overall argument.

Firstly, the study is well-structured, and the performance gains reported are impressive. The study uses well-studied evaluation metrics, such as BERTScore, BLEU, and ROUGE. The study compares against relevant baselines like MemeGuard and MemeMQA. Datasets used are also well-studied in academia and also include some sourced from the internet. While the individual components of the framework (e.g., ICV, semantic similarity-based search, LoRA fine-tuning) are not novel, their combined application for meme detection is.

However, the study focuses on a limited number of models, specifically two models of similar sizes (8B and 9B), which restricts the generalizability of the claim of being a state-of-the-art (SOTA) framework for harmful meme detection. It would be beneficial to clarify that this claim is specific to models of this size and does not extend to larger models like GPT-4, which could potentially perform better in zero-shot or few-shot settings. It would also help to clarify that this might be SOTA only for techniques that don't require finetuning a model and just require post-hoc intervention. Alternatively, authors should consider comparing performance with finetuned model, on zero shot/few shot baselines.

The study also does not include any backbones mentioned in the Memeguard paper, and the reported numbers for Memeguard are significantly different. For instance, the Rouge-L score for Memeguard in this paper is reported as 0.005, whereas the original paper reports its worst performing memguard model(LLAMA) at a score of 5.29. Likewise, Bertscore for the worst performing model has a score of 80.11(LLAMA) but here it is reported as 0.82. Is there a difference in the way authors calculate the metrics or is the disparity because of the backbone chosen(idefics)? If it's the former, I'd encourage the authors to add a section on metrics and explain how they are calculated. If it’s the latter, I’d encourage the authors to add an experiment with the models in memeguard for consistency.

One of the main claims/contributions is the dataset. The dataset is listed as a contribution, but there is no mention of an intent to release it. This should be reworded.

The motivations for choosing different backbones for different components (fintuned Mc and model for in-context intervention) are unclear. It would be beneficial to explain why different models(llava, idefics) or backbones were chosen instead of using the same one.

Error analysis was a nice inclusion in the paper. It would also be helpful to understand the distribution of harmful and non-harmful images in the dataset of 985 memes, as well as the distribution of harmful/non-harmful between the test and train sets. As the error analysis only talks about false negatives but not false positives? Do they not exist at all?

There is also confusion about the dataset used to train the fine-tuned model and the data for the ICV vectors. Are they the same? What is the size and origin of Dc? It's not mentioned at all.

**Requested Changes:**

Please revise main contributions in section 1 if you don't plan to release the dataset.

Please make sure to add all method related details in one section instead of spreading them across multiple sections. Page 4, section 4 redirects reader to section 6. Page 6, section 5, redirects user to section 4.1. Making it hard for readers to follow the dataset prepration. Despite the back and forth,  the size and source of dataset used to finetune LLAVA isn’t mentioned in either of these sections.

It would be helpful to include a section explaining how metrics are defined and computed, as there seems to be a misunderstanding or misreporting of scores like BertScore, which is off by a factor of 100.

How is readability measured? This is not mentioned in the paper,

Symbols mismatch, eqn 1 has alpha, page 7 last line says gamma.

In section 8, in effect the coefficient, alpha should be 0 to be non-trainable and not one? According to eqn 1, scaled vectors are added. So for model to be non trainable shouldn't be 0?

Including an n-shot baseline and fine-tuned baseline(zero shot) for comparison could help in understanding the effectiveness of ICV.

---

> ### Author Response · Authors · 2025-08-17
> **We express our gratitude to reviewer Qozu for insightful and constructive evaluation**
>
> **C1:**  However, the study focuses on a limited number of models...
>
> **Answer:** We appreciate the reviewer’s observation. In this work, we focused on mid-sized open-source models (8B–9B) due to their favorable balance between performance, cost-efficiency, and practical deployability in safety-critical applications. However, our proposed framework is model-agnostic by design and can be readily extended to larger architectures to further enhance intervention quality. We will clarify in the final version of the paper that our SOTA claim is scoped within mid-sized models in post-hoc intervention settings, without requiring full model fine-tuning.
>
> ---
>
> **C2:** The motivations for choosing different backbones...
>
> **Answer:** We appreciate the reviewer’s observation. Our choice of different backbones for different components is intentional and driven by the distinct demands of each stage in the framework.
>
> - For **Stage-1 (commonsense generation):** we use *LLaVA-v1.6-Mistral-7B-hf* because it combines the strong instruction-tuned language capabilities of *Mistral-7B-Instruct-v0.2* with the vision-language grounding of LLaVA, making it particularly well-suited for nuanced commonsense reasoning from multimodal inputs [1].
> - For **Stage-III (final intervention generation):** we employ *Idefics-2-8B/Idefics-9B* and *OpenFlamingo*, as these models have stronger image-anchored representation capabilities and better alignment for multimodal in-context learning tasks [2].
>
> Moreover, our **Cognitive Shift Vector (CSV)** framework is currently implemented and optimized for these backbones (*Idefics-2-8B/Idefics-9B* and *OpenFlamingo*).
>
> Thus, *LLaVA’s strength in text-rich commonsense extraction* complements *Idefics/OpenFlamingo’s strength in image-driven intervention generation*, ensuring optimal performance in each stage of the pipeline. We shall clarify this point in the revised version.
>
> **[1]** Liu, Haotian, et al. *Improved baselines with visual instruction tuning.* Proceedings of the IEEE/CVF conference on computer vision and pattern recognition. 2024.
> **[2]** Laurençon, Hugo, et al. *Obelics: An open web-scale filtered dataset of interleaved image-text documents.* Advances in Neural Information Processing Systems 36 (2023): 71683-71702.
>
> ---
>
> **C3:** It would also be helpful to understand...
>
> **Answer:** The ICMM dataset of 985 memes contains only harmful memes, and our curated dataset also includes only harmful memes (as described in Section 5, *Filtering and Annotation*). Therefore, there are no non-harmful memes in either dataset.
>
> Since our task is not a binary harmful/non-harmful classification, but rather focuses on generating appropriate interventions for harmful content, the standard notion of false positives does not apply.
>
> - **False negatives** are defined as cases where the method generates an intervention that, upon manual inspection, deems the meme safe for posting—thereby failing to address the harm.
> - Since non-harmful memes are absent from the dataset, the concept of false positives does not arise in our evaluation.
>
> **We shall clarify this in more detail in the revised final version of the paper.**
>
> ---
>
> **RC1:** Please revise main contributions in section 1...
>
> **Answer:** We thank the reviewer for pointing this out. We confirm that we **plan to publicly release the dataset** upon acceptance of the paper. We will clearly state it in the revised manuscript, and we will also include details and access instructions in the supplementary material.
>
> A small excerpt of how the data would look like is listed here: [https://anonymous.4open.science/r/MemeSense/Dataset_excerpts.csv](https://anonymous.4open.science/r/MemeSense/Dataset_excerpts.csv).
>
> ---
>
> **RC2:** Please make sure to add all method related details...
>
> **Answer:** We acknowledge this concern and will restructure the manuscript to improve clarity. In particular, we will consolidate all dataset preparation details into **Section 5**, eliminating cross-references to other sections. This will ensure the dataset description is self-contained and easy to follow.
>
> ---
>
> **RC3:** Despite the back and forth...
>
> **Answer:** The dataset contains **484 memes** sourced from diverse social media platforms, with exact links provided in Section 5.
>
> - **Training:** 300 instances are used for Stage 1 and for training the Cognitive Shift Vectors (CSV).
> - In Stage 1, the model is trained using annotated descriptions with commonsense parameters as ground truth.
> - For CSV training, the memes with their commonsense parameters are used as input, and the annotated interventions as output.
>
> The remaining **184 memes** form the test set:
> - 133 with overlaid text
> - 51 without overlaid text
>
> We will explicitly clarify these details in Section 5 to remove any ambiguity about dataset usage and origin.
>
> ---
>
> **Please see the next section for more details (5K limit)**

---

> > ### Author Response · Authors · 2025-08-17
> > **Continuation of previous comment**
> >
> > **RC4:** It would be helpful to include a section explaining...
> >
> > **Answer:** We appreciate the reviewer’s observation and agree that the metric differences merit clarification.
> >
> > - The original *MemeGuard* has two stages where VLM/LLM are involved. As mentioned in Section 6.1 (baselines), we use the same base VLM (*llava-1.6-mistral-7b-hf*) in stage one, and *idefics-2-8b-base* in stage two.
> > - Threshold: We use **0.2** (optimal) vs. **0.5** (in MemeGuard).
> > - OCR: We did not use OCR in our experiments (to adapt for text-free memes).
> > - Reporting: We use exact values (not percentages).
> > - BLEU: We report **average BLEU-4 only** vs. MemeGuard reporting average of BLEU-1, 2, 3, 4.
> >
> > This explains the differences in **Rouge-L** (0.04 vs. 5.29% = 0.0529).
> >
> > **Additional experiments as asked:**
> > 1. Added OCR (slight change: BERTScore 0.79 → 0.80).
> > 2. Replaced *idefics-2-8b-base* with *Flan-t5-large*. Improvements noted:
> >    - BERTScore: 0.79 → 0.82
> >    - SeSS Score: 0.18 → 0.25
> >    - Rouge-L: 0.04 → 0.08
> >    - BLEU-4: 0.001 → 0.009
> >
> > Yet, these results still lag behind **MemeSense** (BERTScore: 0.87, SeSS: 0.31, Rouge-L: 0.11, BLEU-4: 0.009).
> >
> > We will add these findings to Table 4 and clarify metric computation to ensure transparency.
> >
> > ---
> >
> > **RC5:** How is readability measured?...
> >
> > **Answer:** We measure readability using the **Flesch Reading Ease score**, implemented via the *textstat* Python library.
> >
> > This ensures interventions are not only accurate but also coherent and accessible. We will explicitly add this detail in **Evaluation Metrics (Section 6.3)**.  We will refine this further in the final version of the paper.
> >
> > ---
> >
> > **RC6:** Symbols mismatch, eqn 1 has alpha...
> >
> > **Answer:** We thank the reviewer for pointing out the perceived mismatch.
> >
> > - **α (alpha):** learnable coefficient vectors for Cognitive Shift Vectors (CSV), defined in Eqn 1 & Fig 1 (Block 4).
> > - **γ (gamma):** weight factor for second loss term, in Eqn 4 (constant 0.5, Section 6.2).
> >
> > We will sharpen the methodology section to clarify this distinction.
> >
> > ---
> >
> > **RC7:** In section 8, in effect the coefficient...
> >
> > **Answer:** As defined in Eqn 1 & Fig 1 (Block 4), **α (alpha)** controls CSV contribution at each layer.
> >
> > - Setting α = 0 would **remove CSV entirely** (no commonsense adaptation).
> > - In Section 8, fixing α = 1 (non-trainable) means injecting the same CSV at every layer.
> >
> > This allowed us to isolate the effect of uniform injection vs. learnable α. We observed performance degradation compared to trainable α.
> >
> > We will clarify this in the revision.
> >
> > ---
> >
> > **RC8:** Including an n-shot baseline and fine-tuned baseline...
> >
> > **Answer:** We appreciate the suggestion.
> >
> > - Section 6.1 already includes **n-shot baselines** with n ∈ {1, 2, 4, 8, 10} using both random and semantic ICDs.
> > - Additional experiment: fine-tuned *idefics-2-8b* on 300 instances.
> >
> > **Results:**
> > - On memes *without text:* CSV outperformed fine-tuned baseline (SeSS = 0.63, BERTScore = 0.81 vs. 0.71 & 0.91).
> > - On memes *with text:* CSV also superior (SeSS = 0.67, BERTScore = 0.83 vs. 0.78 & 0.91).
> > - On ICMM dataset: CSV again better (SeSS = 0.14, BERTScore = 0.74 vs. 0.31 & 0.87).
> >
> > This indicates **CSV adaptation outperforms direct fine-tuning in low-resource settings**.

---

> ### Author Response · Authors · 2025-08-19
> **Kindly respond**
>
> Dear Reviewer Qozu,
>
> As the author response period draws to a close, we hope that we have adequately addressed your concerns. We would greatly appreciate it if you could kindly acknowledge our response. Please let us know if there are any remaining points you would like us to clarify or expand upon.
>
> Thank you once again for your thoughtful and constructive feedback.
>
> Best regards,
>
> The Authors

---

> > ### Author Response · Authors · 2025-08-24
> > **Kindly respond**
> >
> > Dear Reviewer Qozu,
> >
> > Please respond. We have tried to address your feedbacks.
> >
> > Best
> >
> > Authors

---

### Review · Reviewer_ctrj · 2025-08-05

**Summary Of Contributions:**

This paper introduces MemeSense, an adaptive in-context learning framework for meme moderation that integrates visual and textual understanding with curated commonsense-enriched examples. MemeSense is structured in three main stages: generating human-like commonsense parameters, selecting in-context exemplars relevant to the target meme, and learning cognitive shift vectors to adapt the model's internal representations. By using semantically and socially grounded cues, the framework aims to detect subtle and implicit harms—such as misogyny, vulgarity, or stereotyping—even in memes lacking overt textual content. Experiments on a newly curated dataset and the ICMM benchmark demonstrate MemeSense's superior performance over state-of-the-art baselines, with notable improvements in semantic similarity and BERTScore, especially for text-scarce memes.

**Additional Comments:**

No

**Audience:**

Yes

**Audience Explanation:**

- The paper tackles the challenging and socially impactful problem of harmful meme moderation, with a focus on implicit/visual cues frequently missed by prior text-centric or unimodal systems. The proposed use of cognitive shift vectors (CSV), in-context multimodal exemplars, and explicit commonsense parameterization is a creative and timely direction for adaptive moderation, bridging gaps left by large VLMs.
- The methodology is comprehensively described, with clear mathematical formalization (see Section 4.3), sound integration of loss components, and a relevant approach for semantic retrieval and parameter selection. The learning of CSVs over multiple layers introduces adaptive modulation, offering a smart alternative to simply stacking demonstrations.

**Claims And Evidence:**

Yes

**Claims Explanation:**

- Rigorous experimentation is presented using both a newly curated dataset and the public ICMM dataset. Multiple baselines are compared, including MemeGuard, MemeMQA, Commonsense-anchored prompting, and several k-shot ICL variants. Ablation studies specifically dissect the role of commonsense information and the CSV components (Tables 6–8), and error analysis is performed (Section 8).
- Table 2 and Table 3 show substantial improvements in BERTScore and semantic similarity over previous approaches, particularly for text-scarce memes, supporting claims that the proposed method generalizes across modalities.

**Requested Changes:**

- Positioning and Related Work Completeness: The related work section is thorough regarding immediate predecessors (MemeGuard, MemeMQA, VLM adaptation), but it omits recent, directly comparable systems from 2025 that advance explainable and resource-efficient meme moderation with large multimodal models:
(1). Hee & Lee (2025). This paper describes IntMeme, offering interpretability methods that should be compared or referenced when discussing explainable intervention (belongs in Section 2 and the experiment section) and (2). Pan, Luu, Wu (2025). U-CoT+ addresses decoupled meme understanding, which is highly relevant for MemeSense's semantic retrieval and cognitive shift strategy (could be discussed as an alternative approach in Section 2 or 4, and ideally included in experimental baselines if resource permits).

- Ablation and Robustness Gaps: While several ablations are provided, further explanation or exploration would be valuable: The effect of increasing the number of in-context exemplars (beyond the reported k values) on generalization and overfitting risk is not deeply analyzed. Sensitivity analysis is present, but it would help to dissect common failure modes with more qualitative error cases for culturally ambiguous memes, as revealed in the detailed analysis on Page 10.

Hee & Lee (2025): Demystifying Hateful Content: Leveraging Large Multimodal Models for Hateful Meme Detection with Explainable Decisions. https://arxiv.org/abs/2502.11073

Pan, Luu, Wu (2025): Detecting Harmful Memes with Decoupled Understanding and Guided CoT Reasoning. https://arxiv.org/abs/2506.08477

---

> ### Author Response · Authors · 2025-08-18
> **We express our gratitude to reviewer ctrj for insightful and constructive evaluation**
>
> **RC1:** Positioning and Related Work Completeness...
>
> **Answer:** Thank you for your suggestion. **We will definitely cite the mentioned papers in the related work**. IntMeme first uses the caption (overlaid text) of the meme and **passes through a frozen LMM to generate an interpretation**. Then using the representation of the interpretation along with visual information **it trains a classifier to predict whether the meme is hateful or not**. **Since our task is beyond binary classification, it cannot be directly compared with the IntMeme**. However, we compare the generated interpretation with our generated commonsense. **We observe that it fails to identify the nuanced harm involved in the memes without texts**.
> > Some qualitative examples of the comparison are given in the anonymous link: https://anonymous.4open.science/r/MemeSense/harmful_meme_results%20-%20U-COT_IntMeme%20baselines.csv.
>
> In some cases, for memes with texts, it partially understands the memes, but fails in the overall scenario. In the revised version, we shall add a few qualitative comparisons of the generated commonsense, along with the generated interpretations from IntMeme.
>
> We also try the U-COT+ to generate an explanation with the meme. **Due to decoupled understanding, this method generates a brief explanation in a zero-shot setting. However, U-CoT+ relies on multiple LLM calls, handcrafted task specific guidelines, which limits its applicability to real-time settings**. In contrast, **MemeSense is designed for scalability, data efficiency, and adaptability by leveraging cognitive shift vectors that require only minimal training and perform well even for nuanced memes with limited or no textual cues**. We will provide an illustrative comparison in the Appendix to support this analysis and discuss its tradeoffs.
>
> **RC2:** Ablation and Robustness Gaps...
>
> **Answer:** We thank the reviewer for raising the important point regarding the effect of increasing in-context examples (k-shot) on generalization and overfitting. To address this, **we conducted a focused qualitative analysis using representative memes from each of our three datasets: *memes without text*, *memes with overlaid text*, and *the ICMM dataset***.
>
> For each meme, we provide the same commonsense parameters and generate interventions from the CSV by training with k (= 1, 2, 4, 8, 10) in-context demonstrations in the inference stage. **We observe a clear pattern of semantic refinement in the generated intervention with increasing k (See qualitative examples in the anonymous link:** https://anonymous.4open.science/r/MemeSense/Interventions_with_Varying_k.csv ). **Specifically, the interventions become more contextually aware and nuanced, demonstrating improved reasoning over the commonsense cues**. However, this improvement saturates around k = 8, after which we observe marginal or no gains. This indicates that while additional examples improve generation quality up to a point, beyond that, the benefit plateaus—suggesting minimal risk of overfitting in our framework under low-data conditions. We will add the Table with examples in the Appendix in the final version.
>
> To address your concern regarding more qualitative error cases, we will expand this analysis by adding a detailed breakdown of the six specific failure cases (as mentioned in Section 8) we have identified. **For each case, we will present the exact generated commonsense outputs and the corresponding interventions in a tabular format similar to Table 13 in the Appendix.**
> For example, **in one such case, the generated commonsense highlighted a potential cultural sensitivity (“hand gesture could be misinterpreted as offensive in some contexts”), but the final prediction incorrectly indicated that the meme did not appear to be harmful**. Including these additional examples will make it easier to understand scenarios where the cognitive shift vector or in-context example fails in isolation, which can provide a better insight into the boundaries and limitations of our method. Other examples are added in this anonymous link: https://anonymous.4open.science/r/MemeSense/harmful_meme_results%20-%20Error%20Cases.csv. We will update these results in the Appendix.

---

> > ### Author Response · Authors · 2025-08-19
> > **Kindly respond**
> >
> > Dear Reviewer ctrj,
> >
> > As the author response period draws to a close, we hope that we have adequately addressed your concerns. We would greatly appreciate it if you could kindly acknowledge our response. Please let us know if there are any remaining points you would like us to clarify or expand upon.
> >
> > Thank you once again for your thoughtful and constructive feedback.
> >
> > Best regards,
> >
> > The Authors

---

> > > ### Comment · Reviewer_ctrj · 2025-08-23
> > > **Thanks for the response**
> > >
> > > Thank you for the detailed response. My concerns have been properly addressed.

---

### Decision · Action_Editor_ZGej · 2025-09-06

**Recommendation:** Accept with minor revision

**Audience:**

Yes

**Audience Explanation:**

Although the novelty of this work is somewhat limited, its target problem seems interesting in real-world applications.

**Claims And Evidence:**

Yes

**Claims Explanation:**

All reviewers are mostly positive at the accuracy of this work in terms of both methodology and experiments.

However, the reviewers recommend that the following points should be reflected in the revision.

1. It doesn’t test across a wide range of model sizes or families. The authors agreed to adjust their claim to make clear that their results are specific to mid-sized models (8B,9B) in post-hoc settings (no fine-tuning) unlike their previous claims. They also confirmed they will release the dataset, which wasn’t certain in the original draft.

2. The authors should improve the writing. Some details on metrics are missing, symbols change inconsistently, and the paper jumps back and forth between sections.

3. The MemeSense framework proposed in this paper demonstrates some innovation in identifying multimodal harmful content, but it still suffers from deficiencies in cross-cultural generalization, failure case analysis, interpretability, and practical deployability. Although the authors provide additional experiments and manual evaluation, the evidence is insufficient to support their broader claims.